# Interstellar ices as carriers of supernova material to the early solar system

Martin Bizzarro [1,2] ✉, Martin Schiller [1], Jesper Holst[1], Laura Bouvier[1],
Miroslav Groen[1], Frédéric Moynier [2], Elishevah M. M. E. van Kooten [1],
Maria Schönbächler [3], Troels Haugbølle [4], Darach Watson [4],
Anders Johansen [1,5], James N. Connelly[1] & Emil Bizzarro[1]

Planetary materials show systematic variations in their nucleosynthetic isotope compositions that resonate with orbital distance. The origin of this pattern remains debated, limiting how these isotopic signatures can be used to trace the precursors of terrestrial planets. Here we test the hypothesis that interstellar ices carried supernova-produced nuclides by searching for a supernova nucleosynthetic fingerprint in aqueous alteration minerals from carbonaceous and non-carbonaceous chondrite meteorites. We focus on zirconium, a refractory element that includes the neutron-rich isotope $^{96}Zr$ formed in core-collapse supernovae. Leaching experiments reveal extreme $^{96}Zr$ enrichments in alteration minerals, showing that they incorporated supernova material hosted in interstellar ices. We show that the Solar System's zirconium isotope variability reflects mixing between these ices and an ice-free rocky component. Finally, the presence of supernova nuclides in a volatile carrier supports models where the Solar System's nucleosynthetic variability was imparted by thermal processing of material in the protoplanetary disk and during planetary accretion.

The Solar System's rocky bodies record nucleosynthetic isotope variability[1] reflecting a heterogeneous mixture of stardust from different stellar sources in the protoplanetary disk, including asymptotic giant branch (AGB) stars (the site of s–process nucleosynthesis) and from supernova environments[2,3]. This isotopically anomalous stardust is preserved in primitive chondrite meteorites as presolar grains like silicon carbide (SiC), silicate, diamond, and graphite[4]. As nucleosynthetic signatures are used to infer genetic relationships between asteroidal and planetary bodies, including Earth's precursor material[5–9], elucidating the nature of the carrier phases of nucleosynthetic anomalies is a key and timely objective.

A critical cosmochemical observation is the existence of variability in the abundance of supernovae nuclides in planetary and asteroidal bodies, which at face value appears to resonate with orbital distance[10]. Volatile-poor inner Solar System bodies record depletion in supernova nuclides, whereas water-rich outer Solar System objects exhibit enrichments. Two contrasting, though not necessarily mutually exclusive, classes of models have been put forward to explain the Solar System's nucleosynthetic variability and the isotopic contrast between inner and outer disk bodies. In one model, the nucleosynthetic isotope contrast between inner and outer disk bodies is ascribed to a compositional change of envelope material accreting to the protoplanetary disk in concert with outward transport of early accreted material by viscous disk expansion[11]. The isotopic composition of the early accreted material has been hypothesized to match that of the first Solar System solids, namely calcium–aluminium-rich inclusions (CAIs). In this model, disk expansion leads to the outward transport of CAI-like material, resulting in an isotopically CAI-like composition for the

[1]Center for Star and Planet Formation, Globe Institute, University of Copenhagen, Copenhagen, Denmark. [2]Institut de Physique du Globe de Paris, Université de Paris Cité, Paris, France. [3]Institute for Geochemistry and Petrology, ETH Zürich, Zürich, Switzerland. [4]Center for Star and Planet Formation, Niels Bohr Institute, University of Copenhagen, Copenhagen, Denmark. [5]Lund Observatory, Department of Astronomy and Theoretical Physics, Lund University, Lund, Sweden. ✉e-mail: bizzarro@sund.ku.dk

outermost disk. The inner disk is then replenished by the subsequent addition of material with a distinct inner Solar System composition. Consequently, this model predicts a CAI-like nucleosynthetic isotope signature for the outermost disk, including the comet-forming region. The recent discovery of primitive material inferred to be from the comet-forming region, preserved in a pristine meteorite, offers a means to test this prediction. Van Kooten et al.[12] analyzed the Si, Mg, Fe, and Cr nucleosynthetic compositions of several outer disk clasts and showed that their isotopic signatures differ markedly from those of CAIs. These results are therefore in tension with models that invoke a CAI-like isotopic composition of early accreted envelope materials.

The alternative view is that the Solar System's nucleosynthetic variability, including the isotopic contrast between inner and outer disk bodies, originates from volatility-driven processes in the protoplanetary disk[13,14]. In this view, higher temperatures of the inner disk regions at early times resulted in selective destruction and, thus, depletion of a carrier of supernova nuclides in the precursor material to inner disk bodies relative to the solar composition. However, no carrier with non-refractory thermal properties consistent with that necessary for destruction by disk processes has yet been identified in meteorites. Moreover, most presolar grains in primitive chondrites originate from AGB stars with relatively minor contributions from supernova environments[4]. Thus, identifying a labile carrier of supernova nuclides susceptible to thermal processing by disk and/or planetary processes that explains the observed variability would represent a major step forward.

A large fraction of the dust produced by supernovae in the interstellar medium (ISM) is destroyed by the shock generated by the interaction of the supernova blast wave with its surrounding medium[15]. This shock results in the atomization of the supernova dust into a hot and diffuse cloud phase. Furthermore, the forward shock from the supernovae will destroy dust grains already present in the ISM[16]. As this hot phase expands into the cooler and denser cloud cores, theoretical considerations indicate that atomized nuclides can be incorporated into interstellar ices mantling pre-existing grains[17,18]. Additionally, supernova nuclides can be accelerated and implanted into icy mantles by shock waves in ISM[19]. This predicts that an appreciable amount of supernova nuclides together with reprocessed ISM material may reside as atomic species in volatile interstellar ices rather than in refractory silicate dust grains. The discovery of atomic Fe and Ni in cometary atmospheres interpreted to reflect low temperature sublimation of a volatile component aligns with this model[20,21].

To evaluate the hypothesis that interstellar ices host supernova nuclides, we conducted a systematic search for the signature of supernova nuclides in various types of chondrites, including both carbonaceous and non-carbonaceous chondrites. In detail, we have investigated CR (NWA7655) CM (Murchison and Maribo), CV (Leoville), Tagish Lake, ordinary (NWA 5697) and Rumuruti (NWA 753) chondrites. These objects accreted variable amounts of ices, as indicated by the pervasive aqueous alteration they experienced on their parent bodies[22,23]. The secondary minerals produced by aqueous alteration, namely the interaction of water/ice with primary mineral phases, include carbonates, sulfides, oxides, and clay minerals[24]. Thus, strategically-designed step leaching experiments using mild acids aimed at exclusively dissolving the labile aqueous alteration minerals can be used to determine the isotopic signal of the ice component[25]. We focus on the refractory and immobile element zirconium, which owes its nucleosynthesis to the $s$−process as well as various supernovae-related production sites, often referred to the $r$−process[26] for simplification. In particular, the neutron-rich $^{96}$Zr nuclide is overproduced in the explosive He-burning shells of core-collapse supernovae[27]. Large $^{96}$Zr enrichments are also observed in X-type SiC grains and some presolar graphites, which are both inferred to have a supernova origin[28–30]. Thus, $^{96}$Zr enrichments in meteoritic components are a hallmark of a supernova signature. In contrast to earlier

studies[25,31,32], which employed multiple acid types of varying strength to resolve the isotopic signatures of diverse mineralogical components, we aim to determine the zirconium isotope composition of the ice and ice-free components. For each chondrite, we first apply mild acid leaching to extract a fraction representing the ice-derived component, followed by complete digestion of the residue to determine the composition of the ice-free component.

## Results and discussion
### A volatile carrier of anomalous $^{96}$Zr in chondrites
Figure 1 shows the mass-independent zirconium isotopic compositions of the leach fractions and the residue for each chondrite sample. The data are reported in the $\mu^x$Zr notation (x denoting 91, 92 or 96), which represent the p.p.m. deviations from the terrestrial composition. The leachates record extreme $\mu^{96}$Zr enrichments of up to ~5000 ppm, whereas the residues record $\mu^{96}$Zr deficits down to ca. −900 ppm. This implies that a component with a highly anomalous $\mu^{96}$Zr composition is preserved in a labile mineralogical phase soluble in the mild acid utilized in our leaching procedure. A striking feature of the data is the strong correlated variability of $\mu^{96}$Zr with $\mu^{91}$Zr and $\mu^{92}$Zr. Thus, the Zr isotope variability can be ascribed to the same nucleosynthetic component across the suite of chondrites investigated, which includes objects derived from parent bodies accreted in both the inner and outer Solar System.

As presolar SiC grains are a known important reservoir of anomalous Zr[33], we evaluate their role in producing the observed Zr isotope variability. The main source of SiC is $s$−process enriched mainstream grains[4], characterized by large deficits in $\mu^{91}$Zr and $\mu^{96}$Zr. Thus, the $\mu^{91}$Zr and $\mu^{96}$Zr excesses in the leachate fractions may represent the composition complementary to the SiC-enriched residues. Assuming a SiC abundance of 15 ppm in CM chondrites[34,35] and an average Zr concentration of 90 ppm[36], the effect of not dissolving SiC only produces a 250 ppm excess in $^{96}$Zr, about 20-fold lower than that needed to explain the observed CM chondrite excesses. Rare presolar grains like X-type SiC and presolar graphite are characterized by large $\mu^{96}$Zr excesses. Selective dissolution of these phases could theoretically explain the $\mu^{96}$Zr excesses in the leachate fractions. However, neither SiC nor graphite are expected to dissolve in the weak acid used here for leaching. Additionally, reproducing the ~5000 ppm $\mu^{96}$Zr excess observed in Tagish Lake (ungrouped), Maribo (CM), and Murchison (CM) would require an implausible >10-fold enrichment of these phases in the leachate fractions. Aqueous alteration could, in principle, destroy fragile phases like minute presolar silicates such that the $^{96}$Zr excess could reflect the composition of these grains. However, the fact that Maribo, Murchison and Tagish Lake, which record contrasting degrees of aqueous alteration (from type 2.0 to >2.9[37,38]), have identical coupled $^{96}$Zr-$^{91}$Zr excesses does not support this hypothesis. Thus, we rule out variable contributions from refractory presolar SiC, graphite, or silicate grains as the cause of Zr isotope variability between leachates and residues.

The amount of Zr in the leachate fractions, ranging from 0.4 to 19% of the total Zr budget (Table S1), correlates with the matrix abundance in the meteorites (Fig. S1). This suggests that leachable Zr is hosted by a labile, secondary alteration phase in the matrix that is soluble in weak acid. The acetic acid used in the step-leaching procedure dissolves carbonates, sulfates, and sulfides. Although silicate phases are not soluble in acetic acid, experiments have demonstrated that weak acid can remove cations like Mg, Fe, and Al from the octahedral and tetrahedral sites of phyllosilicates[39–41]. The amount of Zr recovered in the leachate fractions covaries with the relative abundance of Mg, Al, and Fe in the same fractions (Fig. S2), suggesting that the anomalous Zr comes from phyllosilicates. The lack of covariation with the Ca abundance (Fig. S2) confirms that carbonates are not an important host of Zr. Although no Zr concentration data exist for phyllosilicates of the chondrites investigated here, phyllosilicates from

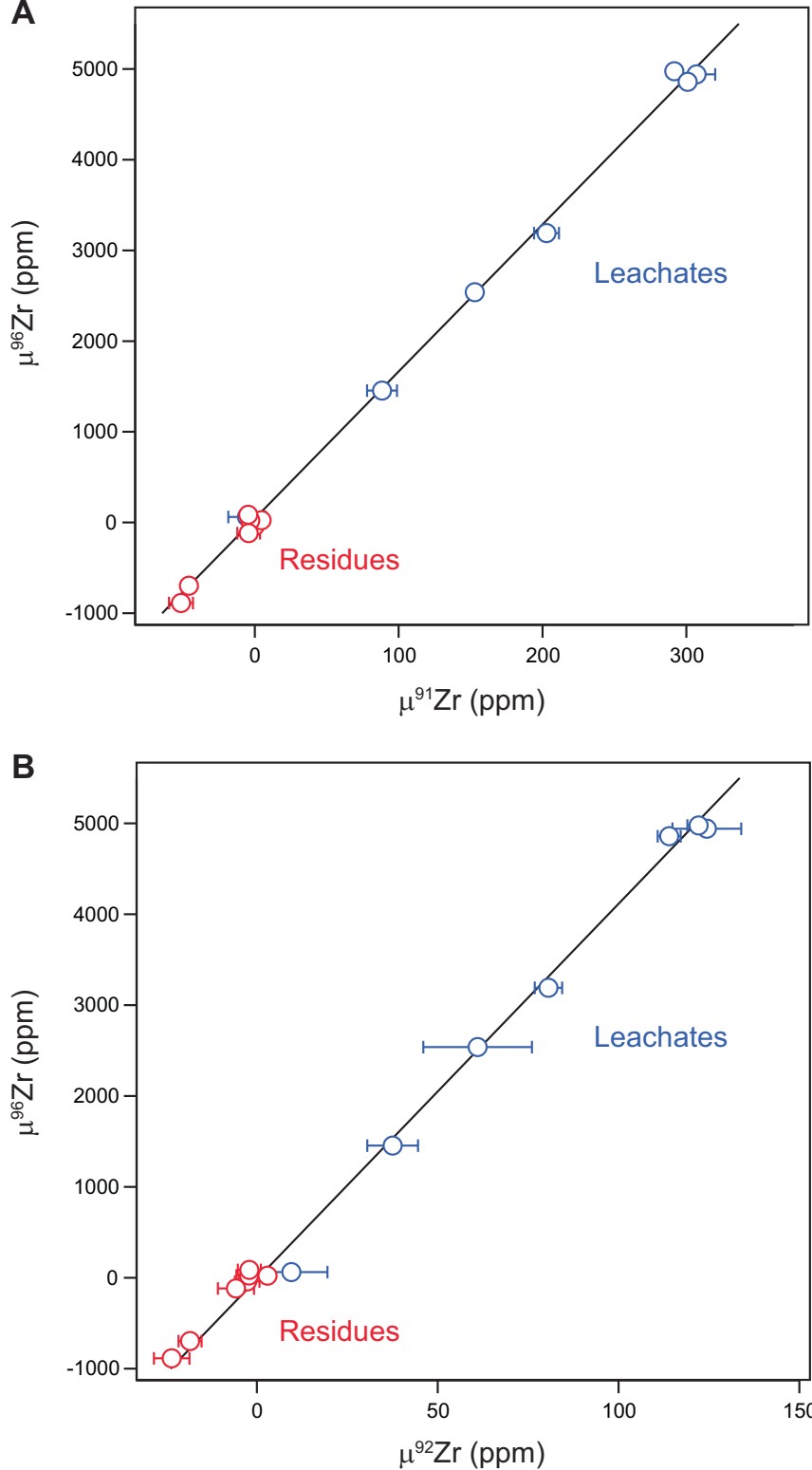

**Fig. 1 | Zr isotope variation diagrams for residues and leachates. A** $\mu^{96}$Zr versus $\mu^{91}$Zr values. The data define a slope of 16.2 ± 0.1 and intercept of 47 ± 18 **B** $\mu^{96}$Zr versus $\mu^{92}$Zr values. The data define a slope of 40.7 ± 0.9 and intercept of 52 ± 54. Uncertainties reflect the internal precision of the measurements (2SE) and are, in some cases smaller than symbols. Note that regressions were calculated using either the quoted 2SE uncertainties or the external reproducibility, whichever was larger. The slopes were report here are identical within uncertainties to those reported in previous leaching experiments[31,32].

the Ryugu asteroid contain 8 ± 1 to 13 ± 4 ppm Zr[42], indicating that this phase can host Zr. In contrast, Ryugu carbonates have negligible amounts of Zr[42]. Thus, we conclude that the anomalous Zr in the leachate fractions is predominantly hosted by phyllosilicates. Since phyllosilicates are hydrous and form through the interaction of water/ice with preexisting silicates, the Zr they contain should represent a mixture of the isotopic composition of the host silicates and that carried by the water/ice. If the anomalous Zr derived solely from the

preexisting silicates, the leachate compositions would be indistinguishable from the bulk chondrite. Instead, the leachates record much more anomalous signatures, demonstrating that the Zr isotope variability reflects a significant contribution from the water/ice component. These findings support a model in which supernova-derived atoms, transported in interstellar ice, were selectively incorporated into phyllosilicates during aqueous alteration on meteorite parent bodies, revealing a volatile carrier of nucleosynthetic anomalies distinct from previously studied refractory grains.

The samples that experienced the lowest peak thermal alteration temperatures (hereafter referred to as the most pristine) are the two CM (Maribo and Murchison) and the Tagish Lake carbonaceous chondrites, which record the highest anomalous Zr isotope compositions in their leachates with nearly identical $\mu^{91}Zr$, $\mu^{92}Zr$ and $\mu^{96}Zr$ values of $299 \pm 2$ ppm, $119 \pm 5$ ppm and $4925 \pm 57$ ppm, respectively, in agreement with earlier work[25]. Although Maribo, Murchison, and Tagish Lake have similar matrix abundances, the amount of Zr recovered from Tagish Lake is at least half that of Maribo and Murchison, indicating that some anomalous Zr in Tagish Lake resides in phases less susceptible to the acid used in leaching. This is consistent with the $\mu^{96}Zr$ value of Tagish Lake's residue being much less negative ($-117 \pm 12$ ppm) than that of Maribo and Murchison ($-696 \pm 14$ ppm and $-886 \pm 17$ ppm, respectively), such that a fraction of the anomalous Zr was not extracted by the leaching experiments. The phyllosilicates in Tagish Lake are primarily serpentine and saponite, compared to serpentine and cronstedtite in CM chondrites[24,38], suggesting that saponite is more resilient to leaching relative to cronstedtite. The nearly identical $\mu^{91}Zr$, $\mu^{92}Zr$, and $\mu^{96}Zr$ compositions of leachates of Murchison, Maribo and Tagish Lake could represent the pure anomalous endmember composition. However, because phyllosilicates formed through water interacting with preexisting minerals, the Zr isotope compositions of these leachates must reflect a mixture between the ice and rock composition and, hence, represents a minimum estimate for the Zr isotope composition of the anomalous interstellar ice endmember. Leachates from other chondrites show more variable and less anomalous Zr isotope compositions than Tagish Lake and the CMs, a pattern also observed in the residues. Given that the $^{96}Zr/^{91}Zr$ and $^{96}Zr/^{92}Zr$ ratios of leachates and residues are invariant across all chondrites, it is unlikely that the anomalous Zr endmember in Tagish Lake and the CMs differs from the other chondrites. Parent body thermal metamorphism can modify the original characteristics of meteorites and, hence, progressive homogenization of the primary isotopic signal of the various components. Apart from Tagish Lake and the CMs, which have not been heated above 150 °C[43], all other meteorites studied have experienced metamorphism above 250 °C[44]. Thus, the decreasing $\mu^{96}Zr$ values in Leoville (CV3.1), NWA 7655 (CR2), NWA 5697 (L3.15), and NWA 753 (R3.9) likely reflect progressive homogenization of the anomalous endmember with the residue. This is supported by the least anomalous composition being found in NWA 753, which experienced the most thermal metamorphism based on its petrological type (3.9).

## Nucleosynthetic source and origin of the Solar System $^{96}Zr$ heterogeneity

Nucleosynthesis of the $^{96}Zr$ nuclide has been linked to the neutron burst triggered by the supernova shock passage through the He shells of core-collapse supernovae[25,45]. The Zr isotope compositions of rare X-type SiC and presolar graphite grains—both characterized by large $^{96}Zr$ excesses—can be qualitatively reproduced by supernova yield models[27], supporting a supernova origin for these grains. Figure 2 compares the average Zr isotopic pattern of leachates from Tagish Lake and CM chondrites with those of X-type SiC, presolar graphite grains, and a core-collapse supernova (CCSN) model. Although the $^{96}Zr$ excesses in supernova-derived grains and CCSN models are ≥100 times larger than those observed in the leachates, this discrepancy likely

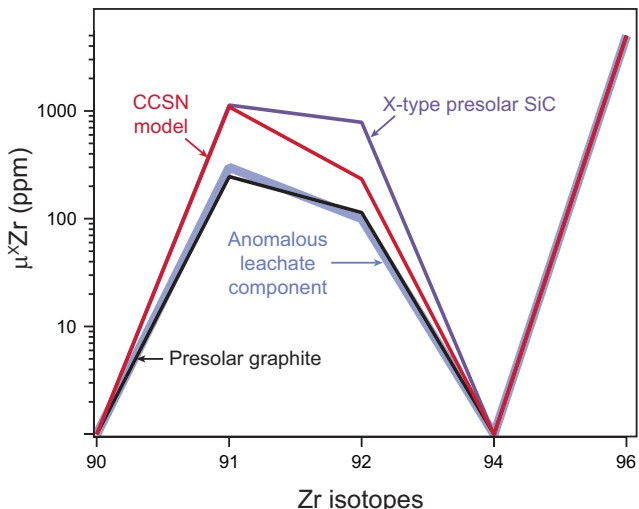

**Fig. 2 | Zr isotope spectrum of anomalous leachate component plotted together with a core-collapse supernova model (CCSN) and presolar graphite and X-type SiC (silicon carbide) grains.** Data are normalized to the $\mu^{96}Zr$ value of the anomalous leachate component. Presolar graphite data correspond to the average of grains 5.01 and C.14 from ref. [28]. whereas the X-type SiC data correspond to the average of grains 113-2, 113-3 and B2-05 from ref. [29]. The CCSN model is for a 25 $M_\odot$ star from ref. [27].

reflects dilution of $^{96}Zr$-rich supernova ejecta within the ISM prior to Solar System formation. Importantly, the relative isotopic pattern in the leachates resembles that produced by neutron-capture nucleosynthesis in the CCSN He shell, particularly the pattern recorded in presolar graphite grains, which shows the closest match among known presolar grain types. We note, however, that this match is not exact: deviations are observed for $\mu^{91}Zr$ and $\mu^{92}Zr$, and the presolar grain patterns shown in Fig. 2 represent averages of only a few grains each, potentially underestimating the full diversity of nucleosynthetic signatures present in supernova ejecta. Moreover, comparison with CCSN model predictions must be viewed with caution, as these models are subject to significant uncertainties related to progenitor structure, explosion conditions, and nuclear physics inputs. Nevertheless, the close resemblance in pattern shape between the leachate data, presolar graphite grains, and the CCSN He-shell model supports the interpretation that the leachates preserve a diluted supernova component, plausibly linked to neutron-burst nucleosynthesis in a core-collapse supernova (see Supplementary Information).

Given its refractory nature, the presence of a significant amount of Zr in the leachates suggests that atomic Zr was captured onto ice-covered refractory grains in molecular clouds and, moreover, that tunneling of the Zr through the ice mantle is suppressed as suggested for Si[18]. The diluted supernova signature inferred from our results supports the idea that some of this atomic material originated directly from freshly atomized supernova ejecta. This must represent a new addition of material to the molecular cloud because similar refractory elements, such as Ti, are strongly depleted in the diffuse ISM[46]. The most likely pathway for this is grain sputtering by the reverse shock. The forward shock processes one to two order of magnitude more material containing older ISM dust[47,48], which provides a mechanism to dilute the observed magnitude of the supernova signal in the atomic reservoir. Sputtering of grains by shock waves in molecular clouds may also contribute to atomic Zr, which later re-embeds into the ice[49]. Thus, our results imply that refractory dust formation in the cold and dense ISM is indeed prohibited by grain ice mantles, as suggested by ref. [17]., and that supernovae are significant sources of specifically atomic phase metals in the ISM.

To understand the relationship between the anomalous Zr nucleosynthetic component in the leachates and the Solar System Zr isotope variability, we analyzed the Zr isotope composition of two $^{26}$Al-poor and three $^{26}$Al-rich calcium-aluminium-rich refractory inclusions (CAIs) as well as bulk Mars and differentiated planetesimals such as Vesta and the angrite parent body (APB). The two $^{26}$Al-poor CAIs are FUN (Fractionated and Unidentified Nuclear effects) type inclusions, whereas the $^{26}$Al-rich are normal, canonical, CAIs. We also examined bulk meteorites from both non-carbonaceous and carbonaceous chondrite groups. Our data improve precision for $\mu^{91}$Zr and $\mu^{96}$Zr values by up to a factor of 3 to 5 compared to earlier studies (see Supplementary Information). Typically, $^{26}$Al-poor CAIs show depletions in supernova nuclides like $^{54}$Cr, $^{48}$Ca, and $^{50}$Ti, while $^{26}$Al-rich CAIs record excesses in these nuclides[13,50–52]. As shown in Fig. 3, the $\mu^{91}$Zr and $\mu^{96}$Zr compositions of the $^{26}$Al-poor and $^{26}$Al-rich CAIs fall on the correlation line defined by the leachates and residues, with deficits and excesses relative to the terrestrial composition, respectively.

A striking observation is that CAIs exhibit both the largest $\mu^{96}$Zr excesses and depletions, despite having formed under similarly high-temperature conditions close to the Sun[53]. This duality appears counterintuitive if $\mu^{96}$Zr variability is linked to the selective incorporation interstellar ices, as CAIs are generally considered the most thermally processed Solar System materials. However, the presence of $\mu^{96}$Zr variability in CAIs does not imply that the $^{96}$Zr carriers survived as intact interstellar ice grains in the inner disk. Rather, the $^{96}$Zr atoms originally embedded in interstellar ices would have been released into the gas phase upon sublimation near the snowline. At heliocentric distances of ~1 astronomical unit, dust had grown to large sizes and settled toward the mid-plane, whereas the gas retained a larger vertical scale height[54]. CAIs are understood to form in transient, high-temperature environments above the condensation front, potentially driven by vertical mixing, convection, or local heating[55]. Because vertical diffusion is inefficient on short timescales, proceeding over tens to hundreds of orbits[56], anomalous $^{96}$Zr released from sublimated interstellar ices could remain confined to the upper disk layers, while the mid-plane gas would increasingly be dominated by isotopically normal evaporated dust. In this setting, CAIs condensing from upper-layer gas would inherit positive $\mu^{96}$Zr signatures, whereas those forming closer to the mid-plane would record depletions. Furthermore, only CAIs forming at high altitudes could be efficiently entrained in disk winds and transported outward, increasing the likelihood of their preservation in the meteoritic record. This vertical isotopic stratification provides a natural explanation for the observed spread in $\mu^{96}$Zr among CAIs and supports a model in which $\mu^{96}$Zr variability is inherited from the processing and redistribution of interstellar carriers, rather than direct accretion of ices.

Similarly to CAIs, all planetary and asteroidal materials, as well as bulk chondrites, plot along the correlation line defined by the leachates and residues, falling between the endmember compositions of $^{26}$Al-poor and $^{26}$Al-rich CAIs. This demonstrates that the Solar System Zr isotope heterogeneity reflects variable incorporation of at least one isotopically anomalous Zr phase, which we hypothesize to be hosted in interstellar ices inherited from the proto-solar molecular cloud core. Importantly, the presence of both $^{96}$Zr-enriched and $^{96}$Zr-depleted CAIs demonstrates that significant Zr isotope heterogeneity existed in the CAI-forming region. While such heterogeneity is expected based on prior work on other isotope systems[57], the critical point here is that this isotopic diversity appears to have been established very early in Solar System history. Disk models suggest that CAI formation conditions were confined to the first ~50,000 years of disk evolution[58], a conclusion supported by high-resolution $^{26}$Al–$^{26}$Mg chronometry of $^{26}$Al-rich CAIs[50]. That both $^{96}$Zr-rich and $^{96}$Zr-poor CAIs formed during this brief period implies the contemporaneous presence of isotopically distinct reservoirs. This observation is difficult to reconcile with models invoking gradual, time-dependent changes in the composition

of envelope material accreting to the disk over several $10^5$ years to account for the isotopic diversity contrast between inner and outer disk bodies[11].

## Planet formation pathways and the solar ice-to-rock ratio

The discovery that volatile interstellar ices host supernova-derived nuclides has significant implications for understanding the precursor material to terrestrial planets and planet formation mechanisms. Critically, it establishes that volatility-driven processes during planetary accretion can modulate the final nucleosynthetic composition of planetary bodies. Earth is enriched in $s$–process-sensitive nucleosynthetic tracers like Zr and Mo compared to other planetary and asteroidal bodies. This has been used to argue that an important planetary building block for terrestrial planets is unsampled, $s$–process enriched inner Solar System material[7,59]. However, the new Zr isotope data presented here clearly demonstrate that Earth is not a compositional endmember, alleviating the need for a missing inner Solar System reservoir to explain Earth's Zr isotope composition. Indeed, most residues from the leaching experiments and the $^{26}$Al-poor CAIs show Zr isotope compositions depleted in the anomalous Zr ice component relative to Earth. Recent studies suggest that the amount of ice accreted by asteroidal and planetary bodies influences the distribution of iron between their mantles (oxidized FeO) and cores (metallic Fe) and, thus, their core mass fraction[60]. In this view, first generation planetesimals like Vesta and the APB formed by the streaming instability exterior to the ice-line[61] incorporate a high mass fraction of ices, resulting in an oxidized mantle. In contrast, larger bodies like Earth that owe a significant fraction of their growth to pebble accretion develop a hot planetary envelope early in their growth history[8,62], which limits the accretion of ices and, thus, results in a less oxidized mantle relative to first generation planetesimals. Figure 4 shows that the FeO contents of the mantles of Earth, Mars, Vesta, and the APB correlate with their $\mu^{96}$Zr values, indicating that the Solar System Zr isotope variability is a function of the interstellar ice fraction accreted by asteroids and planets. We conclude that Earth's enrichment in $s$–process sensitive tracers like Zr and Mo is a hallmark of its accretion history, namely pebble accretion rather than collisional accretion of planetesimals. This is because only the pebble accretion process can provide the thermal processing needed to sublimate ices during the accretion and thereby reduce the abundance of supernova-derived $^{96}$Zr in planetary bodies.

Thermal processing of solids with interstellar ices results in the loss of the volatile water-ice component, while refractory metals like Zr residing in the ices are transferred to the silicate fraction. Thus, early accreted inner Solar System planetesimals such as Vesta and the angrite parent bodies may lose much of their volatile inventory through magma ocean degassing but retain their initial $^{96}$Zr content. Similarly, thermal processing of disk solids by, for example, chondrule-forming processes can also decouple volatile and refractory elements in the precursor material. Thus, our results offer insights into the relative mass fraction of interstellar ices accreted by solids that formed first generation planetesimals to account for the observed $\mu^{96}$Zr variability, which ranges from ~35 to ~140 ppm in the parent bodies of differentiated and chondritic meteorites. Inferring the amount of ice accreted requires knowledge of the $\mu^{96}$Zr values and Zr concentration of the ice and ice-free endmembers. The two CM chondrites analyzed here give consistent $\mu^{96}$Zr values of $4918 \pm 118$ ppm and $-791 \pm 190$ ppm for the ice and ice-free endmembers, respectively. Note that the inferred $\mu^{96}$Zr for the ice endmember is a strict minimum. The Zr concentrations of the ice and ice-free endmembers are calculated using mass balance arguments based on the amount of Zr recovered from the L3 and residue fractions for Murchison and Maribo and adopting an ice/rock ratio of 0.25 (by mass) for CM chondrites (see Supplementary Information). Figure 5 shows that the Solar System variability recorded by asteroidal bodies can be reproduced by mixing 22–25% (by mass) of the anomalous component representing

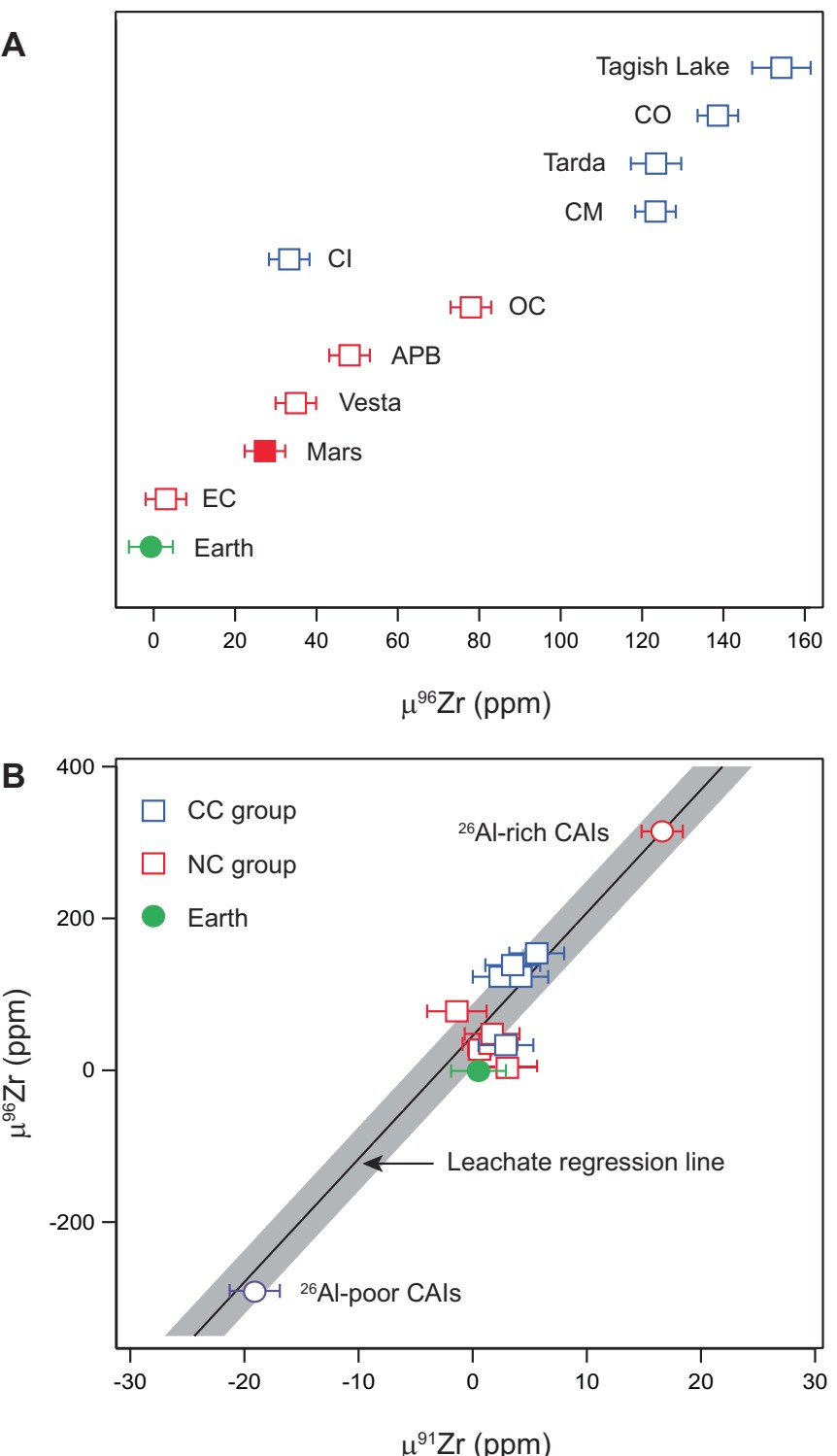

**Fig. 3 | Zr isotope composition of bulk planetary materials as well as $^{26}$Al-rich and $^{26}$Al-poor CAIs. A** $\mu^{96}$Zr of non-carbonaceous (NC) and carbonaceous (CC) parent bodies, including Earth and Mars. **B** $\mu^{91}$Zr- $\mu^{96}$Zr variation diagram for bulk planetary material plotted in A) together with $^{26}$Al-rich and $^{26}$Al-poor CAIs. The uncertainty of the leachate regression line (gray area) represents a 95% confidence interval. CAIs, calcium–aluminium-rich inclusions, OC ordinary chondrite, EC enstatite chondrite, APB angrite parent body, CI Ivuna-type carbonaceous chondrites, CM Mighei-type carbonaceous chondrites, CO Ornans-type carbonaceous chondrites.

interstellar ices with an ice-free rocky component. This narrow range suggests that inner Solar System achondrite and chondrite parent bodies accreted similar amounts of interstellar ices as outer Solar System carbonaceous asteroids. Moreover, the rock-to-ice ratio of ~3 inferred for these bodies is strikingly similar to that of comets and

Kuiper Belt objects[63]. This suggests a common formation pathway for asteroids and comets, highlighting a compositional continuum between these objects[12].

We note that CI chondrites record the least anomalous $\mu^{96}$Zr compositions relative to other carbonaceous chondrites. Thus, the

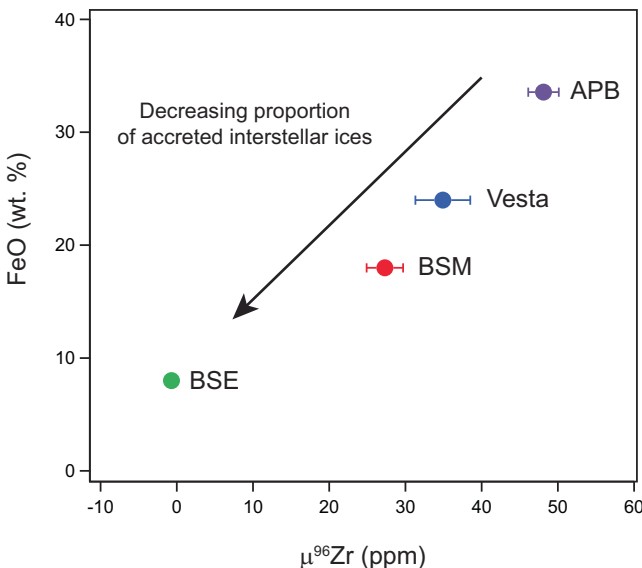

**Fig. 4 | Relative proportion of interstellar ices accreted by differentiated asteroids and planets.** The FeO concentration of the mantles of Earth, Mars, Vesta and the angrite parent body (APB) plotted as a function of their $\mu^{96}Zr$ values. The FeO estimates are from ref. 75. (Earth and Mars), refs. 76,77.(Vesta) and ref. 78 (APB). Uncertainties reflect the internal precision of the measurements (2SE). BSE bulk silicate Earth, BSM bulk silicate Mars.

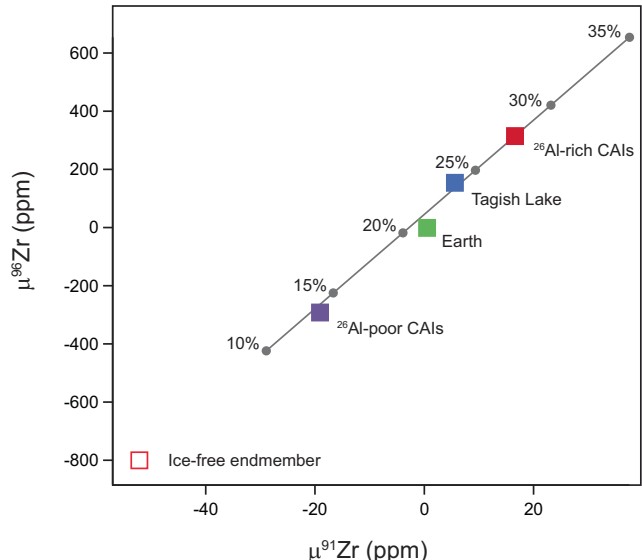

**Fig. 5 | Estimated interstellar ice abundances incorporated by the various solids and/or bodies to reproduce their $\mu^{96}Zr$ values.** The amount of ice accreted to reproduce the variability observed between Earth and Tagish Lake correspond to approximately 22–25% by mass. This calculation assumes that the $\mu^{96}Zr$ ice and ice-free endmembers are 4900 ppm and −800 ppm, and that the Zr concentrations of the two endmembers are 2.29 ppm and 3.6 ppm, respectively. An underestimation of the $\mu^{96}Zr$ value of the anomalous ice endmember by an order of magnitude (and the associated calculated Zr concentration) results in a comparable range of abundances of ice accreted corresponding to 18–22% by mass. We conclude that the narrow range of accreted ices between the bodies inferred here is a robust result. CAIs calcium–aluminium-rich inclusions.

muted $\mu^{96}Zr$ signatures of CIs, despite their status as some of the most water-rich meteorites, indicate that the abundance of accreted water ice and the preservation of isotopically anomalous nuclides inherited from interstellar ices are not necessarily correlated. This

discrepancy can be understood in the context of the accretion environment of CI chondrites relative to other carbonaceous chondrites. Van Kooten et al. [12] recently reported isotopically extreme, organic-rich dark clasts in a CR chondrite, inferred to be of cometary origin based on their chemistry, petrology, and highly anomalous isotopic signatures ($\delta^{15}N$-rich, D-rich, and $^{16}O$-poor). In nucleosynthetic $\mu^{30}Si-\mu^{54}Fe$ space, these clasts define a correlation with carbonaceous chondrites, while a separate correlation links CI chondrites with both non-carbonaceous and carbonaceous bodies. These dual correlations suggest that CI chondrites formed at the dynamic interface between the NC and CC reservoirs, likely near the water snowline and sunward of Jupiter's orbit. In such a region, grains crossing the snowline will undergo repeated sublimation and recondensation cycles driven by radial drift and disk dynamics[64]. When interstellar icy grains drift inward and cross the snowline, their water ice mantles sublimate, releasing not only water vapor but also embedded atomic species such as anomalous $^{96}Zr$. While the water vapor can diffuse outward and recondense beyond the snowline, refractory elements like Zr do not necessarily follow the same path. Once released, these atoms may either condense into mineral phases with different isotopic compositions or become diluted by mixing with isotopically "normal" gas in the inner disk. Consequently, a body like a CI chondrite could still accrete a significant mass of recondensed water ice, yet record only modest $\mu^{96}Zr$ anomalies due to the loss or isotopic homogenization of the original anomalous component. Moreover, the near-absence of CAIs and chondrules in CI chondrites is consistent with formation inside Jupiter's pressure bump, which would have filtered out mm-sized high-temperature components, further supporting a snowline formation scenario for these meteorites. In this view, other carbonaceous chondrites formed beyond Jupiter and were not affected by snowline cycling.

Collectively, our data support models proposing that first-generation planetesimals formed by streaming instability at or beyond the snow line[61,64]. This aligns with findings suggesting that non-carbonaceous, first-generation planetesimals accreted substantial amounts of water-bearing materials[65]. Some chondrite meteorites contain high fractions of high-temperature components like chondrules and refractory inclusions. The similar inferred interstellar ice/rock ratio of chondrite parent bodies with varying amounts of high-temperature components implies thermal processing of early-formed solids or precursors occurred together with interstellar ices. In detail, both $^{26}Al$-free and $^{26}Al$-rich refractory inclusions analyzed here show $^{96}Zr$ and $^{91}Zr$ excesses relative to ice-free end-member compositions defined by the CM residues (Fig. 3). Similarly, one pooled chondrule aliquot from the Allende chondrite records a $\mu^{96}Zr$ value of $125 \pm 45$[26], implying its precursor was thermally processed together with the anomalous $^{96}Zr$ component. Accepting that the $\mu^{96}Zr$ values of $^{26}Al$-poor and $^{26}Al$-rich CAIs represent compositional endmembers of first-generation solids, our data requires that approximately 14% to 27% of the anomalous ice-component was admixed to their precursors. Although the absolute amount of the anomalous ice component admixed to solids and/or bodies inferred here is dependent on the assumed ice fraction accreted by CM chondrites, the overriding observation is that asteroidal bodies and first-generation solids record a narrow range of $\mu^{96}Zr$ compositions relative to ice and ice-free end-members compositions. This requires a relatively constant interstellar ice/rock ratio for the material precursor to asteroidal bodies and, moreover, indicates that thermal processing of primary solids also occurred with similar interstellar ice abundances. This predicts that individual chondrules from different chondrite classes will also record a narrow range of $\mu^{96}Zr$ compositions. Lastly, our results align with models suggesting that a significant fraction of the primary volatile-rich ice reservoir in the early Solar System was directly inherited from the molecular cloud[66] rather than having formed in the protoplanetary disk.

## Methods

### Sample preparation, chemical purification and isotope analysis

Approximately 1.2–1.7 g of each of the chondrites selected for the leaching experiments were sampled and powdered with an agate pestle and mortar. We used a three-step leaching procedure, where the samples were first treated with 10 ml of Milli-Q® (MQ) water for 10 min at room temperature (step L1), 10 ml of 0.4 M acetic acid for 10 min at room temperature (step L2), and, lastly, 10 ml of 8.5 M acetic acid for 24 h at room temperature (step L3). After each step, the sample was centrifuged, and the supernatant transferred to a Savillex™ Teflon beaker for further processing. Residues remaining after individual leaching steps were not dried down before proceeding to the next step. The final residue was fully dissolved by a combination of hot plate and Parr™ bombs digestion using HF-HNO$_3$ mixtures. In detail, the initial hotplate digestion in concentrated HF-HNO$_3$ at 150 °C was followed by dissolution in concentrated aqua regia at 150 °C. All hotplate digestion aliquots were centrifuged in 6 M HCl and the solid residual extracted. The residuals were loaded into Parr™ bomb vessels and fully digested at 210 °C for 2 days following a one-day ramp up step at 150 °C. Subsequently, all samples were fluxed in concentrated aqua regia at 120–150 °C. Full dissolution was achieved in 6 M HCl after which they were dried down in preparation for chemical purification.

For the bulk chondrite samples (SAH 97159, Atlanta, Plainview, Raglan, Roosevelt, SAH 99544, Tagish Lake, Tarda, Murchison and Orgueil), approximately 0.5–1.0 g of material was sampled whereas we extracted approximately 0.1–0.4 g of material for the bulk achondrites (Juvinas, SAH99555, Zagami, NWA 2977, and Nakhla). We extracted one igneous CAI (B12) and one ameboid olivine aggregate (AOA R18) from slabs of the NWA 3118 CV3 chondrite. Approximately 0.06 g and 0.05 g of material was processed for the B12 and AOA R18 inclusions, respectively. The dissolution procedure for the aforementioned material was identical to that of the residue for the leaching experiments. We also processed three previously characterized inclusions, namely the E31CAI[67] from the Efremovka CV3 chondrite, as well as two FUN-type CAIs, namely KT-1[68] and STP-1[52], from the NWA 779 and Allende CV3 chondrites, respectively. For these three inclusions, we used chemistry washes (Ti, Zr, Hf cuts) from earlier chemical purification, and, as such, no digestion was required. In addition to the meteorite samples, we also studied three terrestrial rock standards, namely BHVO-2, BCR-2, and BIR-1. To limit potential sample heterogeneity, we elected to process aliquots from the same large digestion for all three terrestrial standards selected in this study. In detail, we processed amounts of material for the individual aliquots corresponding to between approximately 700 ng and 3500 ng of Zr, which represents the typical amounts of Zr present in the meteorite samples. The Zr purification was achieved with three stages of ion chromatography separation inspired from earlier work[69,70]. The first column involves separation of Al and HFSE (e.g., Ti, Zr, Hf, Mo, and W) from matrix elements using BioRad™ AG50W-x8 cation exchange resin (200-400 mesh). Each aliquot was dried down and redissolved in 0.5 M HCl and loaded on the cation exchanger. Aluminum and HFSE were eluted in 0.5 M HCl-0.15 M HF, followed by the matrix elution in 1.5 M HCl and, finally, REE elution in 6 M HCl. The second column enabled separation of Ti, Al from Zr, Hf, and Mo using Eichrom™ TODGA resin (50–100 μm). After evaporation to dryness, the Al-HFSE fraction was re-dissolved in 0.85 mL of 3.5 M HNO$_3$-0.06 M H$_3$BO$_3$ and loaded on the collum. Aluminum and Ti were washed in 3.5 M HNO$_3$, while Zr and Hf were retained by TODGA resin. Zirconium with Hf and Mo were subsequently collected in 1 M HNO$_3$ + 0.35 M HF. Zirconium ±Mo-Hf cuts were then processed through a final clean-up step to purify Zr from Mo using 55 μL of Biorad™ AG50W-x8 cation exchange resin. Zirconium±Mo cuts were dried down and redissolved in 0.5 mL of 0.5 M HNO$_3$ and loaded on the column. Molybdenum was washed off in 0.5 M HNO$_3$ followed by Zr elution in 0.5 M HCl-0.15 M HF. Zirconium separation from Hf has been deemed to be not necessary to

obtain accurate Zr isotope data using MC-ICP-MS[71]. However, given the <10 ppm analytical precision achieved in this study for $\mu^{96}$Zr, potential isobaric interference from doubly charged Hf warrants careful evaluation. To test this, we conducted doping experiments using solutions with Zr/Hf ratios as low as 10—approximately a factor of three less favorable than the lowest ratios measured in our samples. The resulting $\mu^{96}$Zr values were indistinguishable from those of undoped reference solutions within uncertainty, demonstrating that Hf$^{2+}$ interferences are negligible at the precision of our measurements (see Supplementary Fig. S4). After drying down, the Zr fractions were purified from residual organics from the resin using a mix of 7 M HNO$_3$-H$_2$O$_2$. Zirconium yields from the entire chemical protocol were higher than 85% and assessed from rock standard BIR-1 and BHVO-2 aliquots with similar amount of Zr than for bulk meteorite aliquots. Total procedural chemistry blanks, including sample dissolution and ion exchange separation, were typically below 350 pg of Zr and, therefore, negligible.

High precision Zr isotope analyses were performed using a ThermoScientific™ Neptune Plus™ MC-ICP-MS at the Centre for Star and Planet Formation (University of Copenhagen). Following Zr purification, samples were dissolved in a dilute nitric acid solution containing trace of HF (2% HNO$_3$ + 0.01 M HF) and introduced into the plasma source via an Elemental Scientific Instruments Inc.™ HF Apex™ desolvating nebulizer. All analyses were conducted with Jet sample cone and skimmer X-cone to enhance sensitivity. The typical sensitivity was 400–600 V/ppm at an uptake rate of about 50 μl/min. The five Zr isotopes, $^{90}$Zr, $^{91}$Zr, $^{92}$Zr, $^{94}$Zr, and $^{96}$Zr were measured in static mode using five Faraday cups connected to amplifiers with $10^{11}\Omega$ feedback resistor. In addition, $^{95}$Mo, $^{98}$Mo, and $^{99}$Ru were monitored in Faraday cup connected to amplifiers with $10^{13}$ feedback resistor to correct for isobaric interferences from $^{96}$Mo and $^{96}$Ru on $^{96}$Zr, $^{94}$Mo on $^{94}$Zr, and $^{92}$Mo on $^{92}$Zr. Data were acquired using the sample-standard bracketing technique, namely, one sample measurement was interspaced with one or two measurements of Alfa Aesar™ Zr reference standard solution. The $^{91}$Zr/$^{90}$Zr, $^{92}$Zr/$^{90}$Zr and $^{96}$Zr/$^{90}$Zr were internally normalized to a constant $^{94}$Zr/$^{90}$Zr value of 0.3381[72]. Results are reported in the μ notation, which reflects the ppm deviation of the internally normalized ratios relative to the reference standard. During an analytical session, samples were typically measured up to ten times depending on the amount of Zr available for each sample. Each measurement typically consisted of 100 integrations of 16.78 seconds and was preceded by 1678 seconds of on-peak zero blank in the clean 2% HNO$_3$ + 0.01 M HF solution used to dissolve the samples. Peak centering was performed at the beginning of each standard measurement. Ion beams for the sample and standard solutions were matched within 5%. All solutions were measured at Zr concentrations between 25 and 100 ppb, depending on the total amount of Zr available for each sample. Data reduction was conducted offline using the software package Iolite4. During the data reduction, the entire population of measurements of a 12–24 h analytical session was used to interpolate parameters that varied with time, including instrumental mass bias and background signal. Data reduction consisted of an initial baseline subtraction followed by isobaric interference and instrumental mass fractionation corrections. After rejecting outliers with a 2 standard deviation threshold, the baseline was interpolated using an adequate spline and subtracted to the ion beam intensities of each isotope. Isobaric interference of Ru was then estimated based on the measured intensity on mass 99, using Ru abundances[73] and normalizing to $^{94}$Zr/$^{90}$Zr = 0.3381. Similarly, isobaric interferences of Mo were estimated based on the measured intensity on mass 95, using accepted Mo isotopic abundances[74] and normalizing to $^{94}$Zr/$^{90}$Zr = 0.3381. For each sample, uncertainties are reported as 2SE and reflect the standard error of the mean from multiple replicate measurements. To evaluate potential isobaric interference on mass 96 from polyatomic species, we routinely monitor the production rate of $^{56}$Fe$^{40}$Ar during MC-ICP-MS

analyses. Under our standard operating conditions (using Jet and X cones), a 1 V signal on $^{56}$Fe produces a corresponding signal of ~$10^{-5}$ V at mass 96, corresponding to a production rate of ~10 ppm. Residual Fe concentrations in our purified Zr solutions, measured by quadrupole ICP-MS, yield Fe/Zr ratios typically below 0.001 and often significantly lower. Taken together, the low Fe abundance and low $^{56}$Fe$^{40}$Ar production rate ensure that any contribution to mass 96 is extremely negligible, even at the highest level of precision achieved in this study.

The accuracy and external reproducibility of our method was assessed with the terrestrial rock standard data, which comprised three chemically purified aliquots of BIR-1, four chemically purified aliquots of BHVO-2, and three chemically purified aliquots of BIR-1. The average and 2 SD values of these 10 chemically processed aliquots of terrestrial rock standards yield $\mu^{91}$Zr, $\mu^{92}$Zr, and $\mu^{96}$Zr values of $0.5 \pm 2.6$, $0.7 \pm 3.6$, and $-0.7 \pm 5.7$, respectively.

## Major and selected trace element determination

Prior to chemical purification, aliquots were taken from the L1, L2, and L3 leachates, as well as from the residues, and diluted to appropriate concentrations for elemental analysis. Major and selected trace element abundances were measured on the Thermo Scientific™ iCAP RQ™ ICP-MS at the Centre for Star and Planet Formation. High field strength elements (HFSE) were analyzed using 0.5 M HNO$_3$-0.01 M HF run solution, and all other elements were analyzed in 2% HNO$_3$. Pure element and multi-element reference solutions were used as bracketing standards to allow calculation of the element abundances in the samples, which are reported in Supplementary Data 1. The natural reference material BHVO-2 was processed alongside the unknown samples to assess the accuracy of the method. Based on these experiments, we infer an accuracy of 5% for all data reported in Supplementary Data 1.

## Data availability

All data are available in the main text, the supplementary materials, and in the Supplementary Data 1 file.

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

## Acknowledgements

Funding: Funding for this project was provided by the Villum Foundation (Villum Investigator Grant #54476) and the European Research Council (ERC Advanced grant Agreement 833275—DEEPTIME) to M.B. M.S. acknowledges support from the Carlsberg Foundation (cF20_0209) and the Villum Foundation (00025333). E.V.K. acknowledges support from Villum Foundation (Villum Young Investigator Grant #53024).

## Author contributions

Conceptualization: M.B. and M.S. Methodology: M.B., M.S., J.H., L.B. and M.G. Investigation: M.B., M.S., J.H., L.B., M.G., F.M., E.M.M.E.V.K., Maria S., T.H., D.W., A.J., J.N.C. and E.B. Supervision: M.B. and M.S. Writing—original draft: M.B. Writing—review and editing: M.B., M.S., E.V.K., T.H. and D.W.

## Competing interests

The authors declare no competing interests.
