## [Transparent Peer Review file · Nature Communications]

Interstellar Ices as Carriers of Supernova Material to the Early Solar System

Corresponding Author: Professor Martin Bizzarro

Version 0:

Reviewer comments:

Reviewer #2

(Remarks to the Author)

The manuscript entitled "Interstellar Ices as Carriers of Supernova Material to the Early Solar System" by Bizzarro et al. reports new zirconium isotope data of leaching experiments, bulk meteorites, and refractory inclusions. These data are used to explore the origins of nucleosynthetic isotope heterogeneity in the protoplanetary disk and the carriers involved therein. The data are precise and valuable for the planetary science community. The methods appear solid, although they should be double-checked for potential effects from Hf and Fe interferences (see minor comments).

The paper is well-written and the subject matter is important for our understanding of the early solar system, making the paper suited for publication in Nature communications. However, I am not sure the entirety of the data provides a foundation that consistently supports the interpretations that the authors are presenting. Below, I raise a couple of concerns that the authors need to address before the paper can be published. Depending on whether these concerns can be resolved in a convincing and sound framework, the manuscript could be published after minor revisions, or it may require a more substantive overhaul.

Major comments:

1) The paper advocates for thermal processing having caused the observed nucleosynthetic isotope heterogeneity and is dismissive of competing models that argue for compositional changes in the accreting envelope material over time, which is relatable. Nevertheless, the paper would appear more balanced if the introduction presented an objective view of both models as viable possibilities before dismissing either proposition. Perhaps also worth noting is that the two models are not mutually exclusive, despite often being handled as competing interpretations.

2) Given the paper's interpretation that the measured Zr isotope variability is caused by thermal processing of interstellar ices, I am surprised to see that Orgueil shows the smallest ^{96}Zr excess out of all carbonaceous meteorites investigated. According to Figure 5, this would correspond to a comparatively small fraction of ices accreted, which is in stark contrast to the general perception that CI chondrites are some of the most water-/ice-rich solar system materials we know.

3) In this context, it also seems quite counter-intuitive to me that CAIs show both the highest ^{96}Zr excesses (26Al-rich) and the largest ^{96}Zr depletions (26Al-poor), despite both CAI populations presumably having formed close to the sun, where temperatures must have been high. How do the authors explain the highest abundances of interstellar ices (~30% according to figure 5) in some these most thermally processed solar system materials? Wouldn't one expect all CAIs to be consistently closest to the ice-free endmember (with strong ^{96}Zr depletions)?

4) Perhaps related to the previous comment and the sentence in L223-226: I fail to see the connection between a lack of homogenisation within the CAI forming region (or heterogeneity between different CAI forming regions) and the extrapolation of 26Al heterogeneity in the entire (early) solar system. This conclusion seems particularly questionable if the 26Al-poor and 26Al-rich CAIs reflect chronologically separated populations (e.g. Koop+2016, GCA), but likewise if the isotopic differences between 26Al-poor and 26Al-rich CAIs were indeed caused by thermal processing within a single CAI forming region.

Minor comments:

L22: Many recent studies locate the r-process in neutron star mergers or similar neutron-rich environments (Kasen+2017, Hotokezaka+2018, Perego+2021, Holmbeck+2024), yet the paper mentions only core-collapse supernovae as production site of (the r-process nuclide) ^{96}Zr . Do the authors have information on the significance of neutron star mergers vs. CCSN for the production of ^{96}Zr ? Moreover, since all Zr isotopes have variable contributions from s-process nucleosynthesis (Bisterzo+2014), the measured isotope variability in ^{96}Zr could instead reflect s-process carriers from AGB stars. In fact, Elfers+2020 showed that Hf isotope variations in leaching experiments are more easily reconciled with s-process carriers.

L44-L47: Yes, but the first sentence is vague and could be misunderstood in combination with the second sentence: Ref 10 (Trinquier+2007) first demonstrated a disparity in the abundances of ^{54}Cr between inner and outer solar system materials. Yet, this is not truly a 'disk gradient' as there is no continuum with orbital distance: In fact, the abundances of supernovae nuclides in Fe-peak elements (e.g., ^{54}Cr , ^{50}Ti) decrease with increasing orbital distance among inner solar system materials (Yamakawa+2010) before spiking in outer solar system materials, interrupting this potential trend in isotopes of Fe peak elements. Nuclides of higher mass elements (e.g., Zr, Mo, Ru, ...) seem to show a more continuous trend with orbital distance (e.g., Fischer-Goedde+2017), however, these are for the most part r- and s-process derived isotopes, which don't fit the Trinquier+2007 reference.

L79: Either 'owe' needs to be 'owes' or the first clause should read 'isotopes of Zr'

L85: cite Elfers+2020 and Render+2022 here in addition to the Schoenbaechler reference.

L93: How do the slopes in figure 1 compare to previous Zr isotope investigations of leaching experiments? Elfers+2020 and Render+2022 may have used a more aggressive leaching procedure, however, this still allows for a comparison of the regressions and the carriers causing these isotope variabilities.

L155: Something incorrect in this sentence, presumably the word 'saponite' is superfluous?

L171: rather 'dilution' instead of 'homogenization'?

L186: This statement confuses me. In Figure 2, it appears that presolar graphite (black line) shows the best fit to the leachate data (purple line). Perhaps the patterns are mislabelled, or I am missing something? Do (some of) the other patterns perhaps also fit the leachate data within uncertainty, given that they are partially averaged from 2 or 3 grains? Either way, it would be appreciated if more details were provided regarding these patterns (e.g. in the supplement).

L207: To many people in the community, the term 'FUN inclusions' will be more familiar for these two ^{26}Al -poor inclusions. It might be good to label them as such here and/or elsewhere in the manuscript. The low $m^{92}\text{Zr}$ value of KT-1 seems intriguing, does this CAI require the presence of a different presolar phase?

L210: Looking at Figure S3, the uncertainties for most meteorite groups have indeed been much improved upon. For some samples, however, the precision is comparable or slightly inferior compared to existing data (e.g., Mars, OC). Given that the precision is not consistently $\geq 300\%$ better, this sentence needs slight adjustment.

L217: This sentence needs clarification: The observation that various solar system materials plot along a single regression suggests that the heterogeneous distribution of a single isotopically anomalous phase could explain the observed Zr isotope variability. It does not demonstrate that the Zr would be hosted in interstellar ices or that these would be inherited from the molecular cloud. (For instance, this sentence could read "[...] incorporation of at least one isotopically anomalous Zr phase, which we hypothesize to be hosted in interstellar [...]".)

L219-L223: I am not sure I can follow the logic in this sentence. As a first order observation, the information that individual CAIs show both excesses and depletions in ^{96}Zr simply demonstrates Zr isotope variability in the CAI-forming region(s). The fact that the CAI-forming region(s) was/were not perfectly mixed has been previously established for other elements (e.g., Williams+2016, Davis+2018). I don't see how this lack of homogenisation argues either for or against the model of compositional changes in the accreting envelope material over time, especially if ^{26}Al -poor CAIs would represent a (slightly) older population of CAIs (e.g. Koop+2016) that may or may not have formed in a different region of the early solar system.

L224: 'correlates' is not the most accurate word here; it suggests a continuous relationship between ^{26}Al and $m^{96}\text{Zr}$ (as seen in figures 5 or S1). Instead, the ^{26}Al -rich and the ^{26}Al -poor CAIs and their opposite $m^{96}\text{Zr}$ compositions seem to display two endmember compositions with a separation in between. As outlined in major comment 4, the conclusion in this sentence is questionable either way, but if the authors want to stick with this interpretation, they should rephrase this sentence (e.g. ' ^{26}Al is associated with high $m^{96}\text{Zr}$ '). It would also be helpful to include the $^{26}\text{Al}/^{27}\text{Al}$ ratios of these different CAIs somewhere in the paper so that the reader does not have to search through the cited publications to look up these values.

L236: 'alleviating'

L241-244: Pebble accretion models typically assume ~40% outer solar system material involved in Earth's accretion (Schiller+2018). This works well for isotopes of Fe-peak elements, where Earth plots closer to the carbonaceous chondrite groups compared to Vesta and APB (which allegedly did not grow via pebble accretion). Here, in contrast, the $m^{96}\text{Zr}$ values

of Vesta and APB are closer to carbonaceous chondrites relative to Earth, suggesting a comparatively small fraction of outer solar system material involved in Earth's accretion, counter to expectations of a pebble accretion scenario.

L252: Intuitively, I would expect high-energy collisions of accreting planetesimals to produce at least comparable amounts of heat compared to pebble accretion; could collisional accretion therefore not also provide the necessary thermal processing to sublimate ices?

L266: Ice-free endmember: Given that the more aggressive leaching procedures in both Elfers+2020 and Render+2022 resulted in residues with even larger depletions in ^{96}Zr , are these residues not likely to be more representative of the ice-free endmember? If not, what do the authors hypothesize causes these even more depleted Zr isotopic compositions?

L518: Replace 'content' with concentration

L527: Should read 'm ^{96}Zr of the ice and of the ice-free'

L553: 'Roosevelt'

L557: 'amoeboid olivine'

L583: 'column'

L584: The authors assure that interferences from doubly charged Hf are insignificant and cite a previous Zr isotope study as justification. However, the cited study is from 2004 and reports an average uncertainty of $>100\text{ppm}$ for m ^{96}Zr , whereas the authors here often report $<10\text{ppm}$ uncertainties. Given this stark increase in precision, the authors may want to measure standards doped with variable concentrations of Hf and verify at what threshold doubly charged Hf starts affecting the Zr isotope compositions.

L597: Were measurement solutions checked for residual Fe? Even miniscule amounts of Fe easily compromise isotope measurements of Ru-96 due to interferences from $^{56}\text{Fe}^{40}\text{Ar}$, and the even less abundant isotope Zr-96 should be at least similarly sensitive. This is especially significant given the usage of high-sensitivity Jet and X cones in the present study, which are known to enhance the production of polyatomic species.

L591: 'negligible'

L598: What Zr concentrations were the solutions measured at?

L629: Could clarify that the chemical compositions were measured from aliquots that were taken of the leachates (and that these aliquots were diluted).

Figure 1: The caption reads m ^{91}Zr versus m ^{96}Zr , however, the figure shows m ^{96}Zr (y axis) plotted against m ^{91}Zr (x axis). Significant digits of the intercepts need adjustment.

I regressed the leachate and residue data from Table S1 using the online version of IsoplotR, which resulted in an almost identical slope (16.19 ± 0.12) and intercept (46 ± 17) for the m ^{91}Zr -m ^{96}Zr regression, however the m ^{92}Zr -m ^{96}Zr slope (42.55 ± 0.35) and intercept (2 ± 37) appear to be different. Presumably, this is related to the choice of settings for regressing the data, highlighting that the authors may want to provide more details for how these data were regressed. Finally, the χ^2 of both regressions are rather high at 11 and 5 (i.e., significant scatter in the data beyond analytical uncertainties), which could indicate an analytical artifact in the leachate data or, alternatively, the presence of a second isotopically anomalous carrier involved in generating the Zr isotope heterogeneity?

Figure 2: Is the red pattern (labelled as 'CCSN model') referring to the He-shell of a 25M CCSN (L176-187)? Given that the nucleosynthesis of Zr is sensitive to neutron densities, does this pattern change fundamentally when a 15M CCSN is modelled instead? It would be much appreciated if the supplement would provide a table with values for these different presolar components so that the interested reader can reproduce and compare compositions and slopes.

Figure 4: Caption should read FeO (y axis) as a function of m ^{96}Zr (x axis) as opposed to the other way around (alternatively, you could switch the two axes, so that the vertical axis is consistently m ^{96}Zr throughout the paper). It would be good to clarify here that the data points are the mantles of Earth, Mars, etc. (and use BSE, BSM, etc. for the labels in the figure)

Supplementary:

L23: content -> concentration

L123: In addition to the discrepancies between CO3 chondrites, Akram+2015 also report significantly lower m ^{96}Zr values for the CM chondrite group compared to the present study, as well as isotope heterogeneity between different CM2 specimen. Moreover, Table S1 reports differing m ^{96}Zr anomalies for the three Orgueil digestions (27.8 ± 2.2 , 38.2 ± 2 , and 33.8 ± 3.2), which together yield a χ^2 of 24. Based on these combined observations, it seems that Zr isotopic heterogeneity is a common feature in carbonaceous chondrites, which is consistent with their unequilibrated character and the variable

distribution of isotopically anomalous phases, and this would be particularly evident/problematic if only small amounts of material were dissolved. Looking at the Rendo+2022 reference, these authors list only a single measurement for their Orgueil sample, suggesting they indeed may have dissolved a small amount of that meteorite?

L133: 9 missing in m91Zr

Figure S1 and S2: It appears these correlations would be even better defined if the two NCs (NWA 5697 and NWA 753) were excluded from the regression. Perhaps they could be displayed in a different colour?

Table S1: Perhaps the authors can add a sentence or two to elaborate on how uncertainties were determined. Specifically, the uncertainties for NWA 7655 L3 (two measurements) appear a tad irregular with 2.4ppm m91Zr and 15ppm m92Zr.

Table S1: 'residue' and 'Roosevelt'

Reviewer #3

(Remarks to the Author)

Summary:

The authors conducted Zr isotope leaching and bulk measurements of chondrites to test the hypothesis that interstellar ices carried supernova produced nuclides. The authors present a unique data set, a convincing case that aqueous alteration minerals harbor extreme ⁹⁶Zr enrichments (a supernova signature), and that Zr was carried by interstellar ices. The methods are sound and the work supports the conclusions. The data and conclusions of this manuscript are compelling and will be of interest to the community. I congratulate the authors on a nice piece of work.

Comments:

My comments are minor and can be addressed with clarifying the text. I have no major concerns or issues with the manuscript.

L72. Recommend listing all the sample names followed by their classifications. Listing only the names of Tagish Lake, the ordinary chondrite, and the Rumuruti chondrite seems incomplete.

L114: Recommend providing classifications of these meteorites at first use to make it more accessible. The above discusses CMs, and while Maribo and Murchison are CMs, Tagish Lake is not a CM.

L123: Table 1? Should this be Table S1?

L136–138: "Since phyllosilicates are hydrous and form through the interaction of water/ice with preexisting silicates, the anomalous Zr may reflect the composition of the preexisting silicates rather than the water/ice. If correct, the isotopic composition of the anomalous Zr leached out of the phyllosilicates should match the bulk chondrite composition, which is not observed."

-Agreed, and I agree the anomalous Zr is not from the silicates. However, the leachate should be a mixture of the silicate Zr isotope composition and that carried by the water. This is mentioned in Lines 158–161, but perhaps this statement could be brought up earlier?

L144: "The most pristine samples investigated here are the two CM (Maribo and Murchison) and the Tagish Lake carbonaceous chondrites, which"

-Define what is meant by pristine, this is a relative word that is used to mean many different things (i.e., thermal alteration, aqueous alteration). Do the authors mean pristine regarding peak thermal alteration temperature?

Line 296–298: "This requires a relatively constant water ice/rock ratio for the material precursor to asteroidal bodies and, moreover, indicates that thermal processing of primary solids also occurred with similar interstellar ice abundances."

-Throughout the manuscript, there are instances where the word ice is used, but I think what is meant is the interstellar ice component. This can be clarified in the text. For example, I think the first use of ice in the above sentence is intended to be interstellar-ice/rock ratio, not total ice.

Figure 4. Recommend the figure label should read 'interstellar ices', not just ice. Or was total water/ice meant here?

Supplementary File:

Table S1.

-Roosevelt is misspelled, it should be Roosevelt.

Version 1:

Reviewer comments:

Reviewer #2

(Remarks to the Author)

*Reviewer comments to author:

The authors have effectively addressed previous comments and have included helpful clarifications to accommodate potential concerns, leading to significant improvements in the revised version of the manuscript. I have a few more minor comments (mostly clarifications related to the new paragraph starting in line 43), however, I do not need to see another revision of the manuscript.

Minor comments:

L32: Suggest adding 'primarily' before 'imparted'

L55: Replace 'outer' with 'outermost' to clarify that the comet region is meant and not the CC region

L59: Suggest adding 'inferred to be' before 'from the comet-forming region'

L60: It is mentioned later in the manuscript; however, it may be good to clarify here that these are (not Zr but) combined Mg-Si-Fe-Cr nucleosynthetic isotopic signatures

L62: Yes, however, more accurately: "[...] in tension with models that invoke a CAI-like isotopic composition of early accreted envelope materials."

L218: Something missing in this sentence ("in the leachates"?)

L247: Suggest adding 'the thermal processing' (or 'the selective incorporation') before 'of interstellar ices'

Reviewer #1 (Remarks to the Author):

The manuscript entitled “Interstellar Ices as Carriers of Supernova Material to the Early Solar System” by Bizzarro et al. reports new zirconium isotope data of leaching experiments, bulk meteorites, and refractory inclusions. These data are used to explore the origins of nucleosynthetic isotope heterogeneity in the protoplanetary disk and the carriers involved therein. The data are precise and valuable for the planetary science community. The methods appear solid, although they should be double-checked for potential effects from Hf and Fe interferences (see minor comments).

The paper is well-written and the subject matter is important for our understanding of the early solar system, making the paper suited for publication in Nature communications. However, I am not sure the entirety of the data provides a foundation that consistently supports the interpretations that the authors are presenting. Below, I raise a couple of concerns that the authors need to address before the paper can be published. Depending on whether these concerns can be resolved in a convincing and sound framework, the manuscript could be published after minor revisions, or it may require a more substantive overhaul.

We thank the referee for a thorough review and the constructive nature of the comments, which are clearly aimed at improving our manuscript.

Major comments:

1) The paper advocates for thermal processing having caused the observed nucleosynthetic isotope heterogeneity and is dismissive of competing models that argue for compositional changes in the accreting envelope material over time, which is relatable. Nevertheless, the paper would appear more balanced if the introduction presented an objective view of both models as viable possibilities before dismissing either proposition. Perhaps also worth noting is that the two models are not mutually exclusive, despite often being handled as competing interpretations.

We agree and have now expanded the introduction to include the model of a compositional change in the accreting envelope material over time.

2) Given the paper’s interpretation that the measured Zr isotope variability is caused by thermal processing of interstellar ices, I am surprised to see that Orgueil shows the smallest ^{96}Zr excess out of all carbonaceous meteorites investigated. According to Figure 5, this would correspond to a comparatively small fraction of ices accreted, which is in stark contrast to the general perception that CI chondrites are some of the most water-/ice-rich solar system materials we know.

This is an excellent point raised by the referee and one we have also debated among the co-authors. We agree that CI chondrites are among the most water-rich meteorites. However, the abundance of ice and the preservation of isotopically anomalous nuclides hosted in interstellar

ice need not correlate directly. Van Kooten *et al.* (2024) recently reported dark clasts in a primitive CR chondrite that are inferred to be of cometary origin based on their chemistry, organic-rich petrology, and highly anomalous isotopic signatures ($\delta^{15}\text{N}$ -rich, D-rich, and ^{16}O -poor). When plotted in $\mu^{30}\text{Si}$ – $\mu^{54}\text{Fe}$ nucleosynthetic space, NC bodies, CC bodies and dark clasts reveal two distinct correlations: one between CI and inner disk NC bodies, and another between CI and outer disk CC bodies, including the cometary dark clasts. This observation suggests that the accretion region of CIs was located at the interface between NC and CC reservoirs.

Our working model is therefore that CI chondrites formed close to the water ice line, inward of Jupiter's orbit. In this setting, multiple cycles of sublimation and recondensation of water ice occurred (Drażkowska & Alibert, 2017). As icy grains drift inward across the snowline, their water mantles sublime, releasing both water vapor and atomic species embedded in the ice, such as anomalous ^{96}Zr inherited from the ISM. The vapor then diffuses outward and recondenses onto colder grains beyond the snowline. However, refractory atoms embedded in the ice mantles (like Zr) may not follow the water vapor efficiently: they can be released into the gas and either (a) condense into different mineral phases, or (b) become diluted by mixing with isotopically “normal” disk material. Thus, a body like a CI chondrite could still accrete abundant water ice, but much of the anomalous ^{96}Zr originally trapped in ISM ice mantles would have been lost or diluted during repeated snowline cycling. This reconciles their high water contents with their muted $\mu^{96}\text{Zr}$ anomalies. Importantly, this model does not require thermal processing of the rocky component, but rather of the volatile carriers of anomalous nuclides. Moreover, the scarcity of CAIs and chondrules in CI chondrites is consistent with accretion Sunward of Jupiter's orbit, where the pressure bump would have filtered out mm-sized refractory components, further supporting a snowline formation environment for these meteorites. In this view, other carbonaceous chondrites formed further out in the disk and not affected by snowline cycling. We have added a paragraph to the revised manuscript to clarify this point.

3) In this context, it also seems quite counter-intuitive to me that CAIs show both the highest ^{96}Zr excesses (26Al-rich) and the largest ^{96}Zr depletions (26Al-poor), despite both CAI populations presumably having formed close to the sun, where temperatures must have been high. How do the authors explain the highest abundances of interstellar ices (~30% according to figure 5) in some these most thermally processed solar system materials? Wouldn't one expect all CAIs to be consistently closest to the ice-free endmember (with strong ^{96}Zr depletions)?

We thank the reviewer for highlighting this important point. We agree that CAIs formed inside the silicate condensation front, where temperatures were high (Krot *et al.* 2010). However, the key is that the ^{96}Zr carriers were embedded in ices that sublimated at the snowline and subsequently became part of the disk gas together with refractory nanoparticles and atomic species formerly trapped in the ice matrix. At ~AU distances, dust had grown to large sizes and settled toward the mid-plane, whereas the gas retained a larger vertical scale height

(Guerra-Alvarado et al., 2024). CAI formation likely occurred close to the condensation front but was facilitated by vertical temperature inversions, convection, or turbulence that recycled material above the condensation line (Hsu et al., 2025). Diffusion from the mid-plane to the disk surface occurs over up to hundreds of orbital timescales (Houge et al., 2025), so gas in the upper layers could remain enriched in anomalous ^{96}Zr from sublimated ices, while the mid-plane gas was dominated by isotopically “normal” evaporated dust. Consequently, CAIs condensed from upper-layer gas would record high ^{96}Zr excesses, whereas CAIs condensed near the mid-plane would record depletions. Importantly, only CAIs forming high enough in the disk atmosphere could be entrained in disk winds and survive transport away from the Sun. This explains why we observe both ^{96}Zr -rich and ^{96}Zr -poor CAIs, and why the anomalous, ^{96}Zr -rich CAIs are preferentially sampled. We have clarified this reasoning in the revised manuscript.

4) Perhaps related to the previous comment and the sentence in L223-226: I fail to see the connection between a lack of homogenisation within the CAI forming region (or heterogeneity between different CAI forming regions) and the extrapolation of ^{26}Al heterogeneity in the entire (early) solar system. This conclusion seems particularly questionable if the ^{26}Al -poor and ^{26}Al -rich CAIs reflect chronologically separated populations (e.g. Koop+2016, GCA), but likewise if the isotopic differences between ^{26}Al -poor and ^{26}Al -rich CAIs were indeed caused by thermal processing within a single CAI forming region.

This is a minor point and we agree with the referee. As this sentence is not essential to our results or interpretation, we have opted to remove it from the revised manuscript.

Minor comments:

L22: Many recent studies locate the r-process in neutron star mergers or similar neutron-rich environments (Kasen+2017, Hotokezaka+2018, Perego+2021, Holmbeck+2024), yet the paper mentions only core-collapse supernovae as production site of (the r-process nuclide) ^{96}Zr . Do the authors have information on the significance of neutron star mergers vs. CCSN for the production of ^{96}Zr ? Moreover, since all Zr isotopes have variable contributions from s-process nucleosynthesis (Bisterzo+2014), the measured isotope variability in ^{96}Zr could instead reflect s-process carriers from AGB stars. In fact, Elfers+2020 showed that Hf isotope variations in leaching experiments are more easily reconciled with s-process carriers.

We thank the referee for this thoughtful comment. We fully agree that neutron star mergers (NSMs) are now widely accepted as a dominant site for the main r-process, especially for the production of heavy r-process nuclei ($A > 130$), as supported by observational and theoretical work (e.g., Kasen et al., 2017; Hotokezaka et al., 2018; Perego et al., 2021; Holmbeck et al., 2024). However, ^{96}Zr lies at $A = 96$, within the domain of light r-process or “neutron burst” isotopes, and its nucleosynthetic origin is distinct from that of heavier r-process elements. Specifically, model calculations of NSM nucleosynthesis consistently show that ejecta with

low electron fractions ($Y_e < 0.25$) efficiently produce heavy r-process nuclides ($A > 130$) via fission recycling, but they underproduce or bypass lighter nuclei in the $A \sim 90\text{--}110$ range, including ^{96}Zr (e.g., Wanajo et al., 2014; Eichler et al., 2015; Lippuner & Roberts, 2015). These results suggest that neutron star mergers are not significant sources of ^{96}Zr . In contrast, ^{96}Zr is strongly overproduced in models of neutron burst nucleosynthesis in the He shell of core-collapse supernovae (CCSNe) (e.g., Pignatari et al. 2018). These environments feature a short, intense neutron flux acting on pre-existing seed nuclei and are well-suited to produce isotopes like ^{96}Zr , which are not easily accessed by the main r-process path. Importantly, the Zr isotope patterns observed in presolar X-type SiC and graphite grains, which are inferred to originate from CCSNe, display large ^{96}Zr excesses that are well reproduced by these models as we point out in our manuscript. Thus, the isotopic signature of ^{96}Zr anomalies in our data is most plausibly attributed to a supernova origin.

Regarding the second point, we acknowledge that all Zr isotopes receive some contribution from the s-process, particularly in asymptotic giant branch (AGB) stars (e.g., Bisterzo et al., 2014). However, as noted by Akram et al. (2015), ^{96}Zr has the smallest s-process contribution among the stable Zr isotopes, making it particularly sensitive to contributions from explosive nucleosynthesis environments. In fact, the s-process tends to deplete ^{96}Zr relative to other Zr isotopes. This makes ^{96}Zr uniquely diagnostic of non-s-process sources, especially CCSNe.

Finally, while Elfers et al. (2020) found that Hf isotope anomalies in leachates are best explained by variable s-process contributions, Zr and Hf have distinct nucleosynthetic sensitivities. In our case, the strong ^{96}Zr anomalies in labile components (phyllosilicates) cannot be explained by known s-process carriers like mainstream SiC grains, which instead show ^{96}Zr deficits. Furthermore, the fact that the leachates exhibit ^{96}Zr excesses, not deficits, argues against an s-process origin. We discuss this point in the manuscript, where we rule out AGB-derived grains as the source of the observed anomalies based on mineralogical constraints, isotopic ratios, and abundance estimates.

L44-L47: Yes, but the first sentence is vague and could be misunderstood in combination with the second sentence: Ref 10 (Trinquier+2007) first demonstrated a disparity in the abundances of ^{54}Cr between inner and outer solar system materials. Yet, this is not truly a ‘disk gradient’ as there is no continuum with orbital distance: In fact, the abundances of supernovae nuclides in Fe-peak elements (e.g., ^{54}Cr , ^{50}Ti) decrease with increasing orbital distance among inner solar system materials (Yamakawa+2010) before spiking in outer solar system materials, interrupting this potential trend in isotopes of Fe peak elements. Nuclides of higher mass elements (e.g., Zr, Mo, Ru, ...) seem to show a more continuous trend with orbital distance (e.g., Fischer-Goedde+2017), however, these are for the most part r- and s-process derived isotopes, which don’t fit the Trinquier+2007 reference.

We agree with the referee that the sentence is vague and could be misinterpreted. The main point we want to make is that there is an isotopic contrast between bodies formed in the inner and outer Solar System, which is not strictly a gradient as pointed out by the referee. We have

rephrased the sentence in question but we would like to keep the Trinquier et al. (2007) reference since this is the first paper that demonstrated this fundamental observation.

L79: Either 'owe' needs to be 'owes' or the first clause should read 'isotopes of Zr'

Corrected as "owes".

L85: cite Elfers+2020 and Render+2022 here in addition to the Schoenbaechler reference.

Done.

L93: How do the slopes in figure 1 compare to previous Zr isotope investigations of leaching experiments? Elfers+2020 and Render+2022 may have used a more aggressive leaching procedure, however, this still allows for a comparison of the regressions and the carriers causing these isotope variabilities.

The slopes we report in Fig. 1 are within analytical uncertainties of that produced in earlier studies mentioned by the referee. We added a note to the Fig. 1 caption to make this point clear.

L155: Something incorrect in this sentence, presumably the word 'saponite' is superfluous?

Thanks for spotting this. We have removed the superfluous "saponite".

L171: rather 'dilution' instead of 'homogenization'?

We think homogenization is the better word here.

L186: This statement confuses me. In Figure 2, it appears that presolar graphite (black line) shows the best fit to the leachate data (purple line). Perhaps the patterns are mislabelled, or I am missing something? Do (some of) the other patterns perhaps also fit the leachate data within uncertainty, given that they are partially averaged from 2 or 3 grains? Either way, it would be appreciated if more details were provided regarding these patterns (e.g. in the supplement).

We thank the referee for this helpful observation and agree that the presolar graphite grains provide the closest match to the leachate data among the measured grain types, particularly in terms of the ^{96}Zr anomaly. We have revised the main text to clarify this point and to more explicitly state that the relative isotopic pattern in the leachates most closely resembles that of graphite grains, while acknowledging minor mismatches for other Zr isotopes. We also note that the grain patterns plotted in Figure 2 represent averages of a few grains, and may not fully capture the isotopic diversity of individual supernova-derived grains. Additionally, we

also acknowledge in the text that the comparison with theoretical CCSN models should be interpreted cautiously, as these models remain subject to significant uncertainties related to explosion dynamics, progenitor structure, and nuclear physics inputs.

L207: To many people in the community, the term ‘FUN inclusions’ will be more familiar for these two ^{26}Al -poor inclusions. It might be good to label them as such here and/or elsewhere in the manuscript. The low $m^{92}\text{Zr}$ value of KT-1 seems intriguing, does this CAI require the presence of a different presolar phase?

Agreed – we know introduce the FUN term. With respect to the low ^{92}Zr value of KT-1, well spotted by the referee. We measured the Nb/Zr ratio of that inclusion and it is subchondritic, namely much lower than that observed in the ^{26}Al -rich CAIs. We interpret the low ^{92}Zr value of KT-1 as retarded ingrowth from ^{92}Nb relative to normal CAIs. While interesting, we believed that this observation and resulting implications is beyond the scope of our paper.

L210: Looking at Figure S3, the uncertainties for most meteorite groups have indeed been much improved upon. For some samples, however, the precision is comparable or slightly inferior compared to existing data (e.g., Mars, OC). Given that the precision is not consistently $\geq 300\%$ better, this sentence needs slight adjustment.

Agreed. We now state that our data represent an improvement of up to a factor of 3-5 relative to earlier studies.

L217: This sentence needs clarification: The observation that various solar system materials plot along a single regression suggests that the heterogeneous distribution of a single isotopically anomalous phase could explain the observed Zr isotope variability. It does not demonstrate that the Zr would be hosted in interstellar ices or that these would be inherited from the molecular cloud. (For instance, this sentence could read “[...] incorporation of at least one isotopically anomalous Zr phase, which we hypothesize to be hosted in interstellar [...]”.)

We agree with the referee and have modified the sentence accordingly.

L219-L223: I am not sure I can follow the logic in this sentence. As a first order observation, the information that individual CAIs show both excesses and depletions in ^{96}Zr simply demonstrates Zr isotope variability in the CAI-forming region(s). The fact that the CAI-forming region(s) was/were not perfectly mixed has been previously established for other elements (e.g., Williams+2016, Davis+2018). I don’t see how this lack of homogenisation argues either for or against the model of compositional changes in the accreting envelope material over time, especially if ^{26}Al -poor CAIs would represent a (slightly) older population of CAIs (e.g. Koop+2016) that may or may not have formed in a different region of the early solar system.

We thank the referee for this thoughtful comment and agree that the presence of ^{96}Zr excesses and depletions in individual CAIs reflects isotopic heterogeneity in the CAI-forming region(s), which has also been established for other isotope systems. Our intention was not to suggest that this heterogeneity alone rules out the model of compositional changes in the accreting envelope (e.g., Nanne et al., 2019), but rather to highlight a timescale inconsistency. Nanne et al. propose that isotopic variability arose from progressive mixing of isotopically normal material into an initially anomalous disk over a timescale of several 10^5 years. However, CAI formation—both ^{26}Al -rich and ^{26}Al -poor—is thought to have occurred very early and over a short interval, likely within the first $\sim 50,000$ years of disk evolution (e.g., Larsen et al., 2011; Woitke et al., 2024). The presence of both ^{96}Zr -enriched and ^{96}Zr -depleted CAIs within this narrow window suggests that isotopically distinct reservoirs coexisted contemporaneously, rather than forming through sequential accretion of new envelope material. We have revised the text to clarify that this observation challenges the temporal evolution aspect of the Nanne et al. model as originally formulated.

L224: ‘correlates’ is not the most accurate word here; it suggests a continuous relationship between ^{26}Al and $m^{96}\text{Zr}$ (as seen in figures 5 or S1). Instead, the ^{26}Al -rich and the ^{26}Al -poor CAIs and their opposite $m^{96}\text{Zr}$ compositions seem to display two endmember compositions with a separation in between. As outlined in major comment 4, the conclusion in this sentence is questionable either way, but if the authors want to stick with this interpretation, they should rephrase this sentence (e.g. ‘ ^{26}Al is associated with high $m^{96}\text{Zr}$ ’). It would also be helpful to include the $^{26}\text{Al}/^{27}\text{Al}$ ratios of these different CAIs somewhere in the paper so that the reader does not have to search through the cited publications to look up these values.

This sentence has been removed as per major comment 4.

L236: ‘alleviating’

Corrected.

L241-244: Pebble accretion models typically assume $\sim 40\%$ outer solar system material involved in Earth’s accretion (Schiller+2018). This works well for isotopes of Fe-peak elements, where Earth plots closer to the carbonaceous chondrite groups compared to Vesta and APB (which allegedly did not grow via pebble accretion). Here, in contrast, the $m^{96}\text{Zr}$ values of Vesta and APB are closer to carbonaceous chondrites relative to Earth, suggesting a comparatively small fraction of outer solar system material involved in Earth’s accretion, counter to expectations of a pebble accretion scenario.

We appreciate this comment and agree that Fe-peak isotopes in Earth are consistent with significant incorporation of outer Solar System material. However, the $\mu^{96}\text{Zr}$ composition of Earth need not follow the same trend. As discussed in our recent review (Bizzarro *et al.*, 2025), a key feature of pebble accretion is that once a growing protoplanet reaches $\sim 2\%$ of Earth’s mass, it develops a hot gaseous envelope that thermally processes incoming material. At this stage, the icy components of inward-drifting pebbles—which we infer host ^{96}Zr —are

efficiently destroyed in the hot envelope before accretion and recycled back to the gaseous disk, while more refractory components, such as those carrying Fe-peak elements, continue to be delivered to the growing planet. This thermal filtering effect naturally leads to a decoupling between ^{96}Zr and Fe-peak isotope signatures. The relatively low $\mu^{96}\text{Zr}$ value of Earth, compared to Vesta and the APB, is therefore a predictable outcome of pebble accretion dynamics and not inconsistent with substantial inward transport of outer Solar System material. We believe that this concept is already in the paper – see this sentence: *In contrast, larger bodies like Earth that owe a significant fraction of their growth to pebble accretion develop a hot planetary envelope early in their growth history^{8,56}, which limits the accretion of ices and, thus, results in a less oxidised mantle relative to first generation planetesimals.*

L252: Intuitively, I would expect high-energy collisions of accreting planetesimals to produce at least comparable amounts of heat compared to pebble accretion; could collisional accretion therefore not also provide the necessary thermal processing to sublimate ices?

The key here is the development of planetary envelopes during pebble accretion, which does not occur in collisional accretion (see previous comment).

L266: Ice-free endmember: Given that the more aggressive leaching procedures in both Elfers+2020 and Render+2022 resulted in residues with even larger depletions in ^{96}Zr , are these residues not likely to be more representative of the ice-free endmember? If not, what do the authors hypothesize causes these even more depleted Zr isotopic compositions?

The zirconium isotopic composition of the residues in both Elfers+2020 and Render+2022 are dominated by mainstream SiC grains, which are refractory material. This is the reason why these fractions record such negative ^{96}Zr composition. As such, these residues cannot be representative of the ice-free endmember.

L518: Replace ‘content’ with concentration

Done.

L527: Should read ‘ ^{96}Zr of the ice and of the ice-free’

Corrected.

L553: ‘Roosevelt’

Corrected.

L557: ‘amoeboid olivine’

Corrected.

L583: ‘column’

Corrected.

L584: The authors assure that interferences from doubly charged Hf are insignificant and cite a previous Zr isotope study as justification. However, the cited study is from 2004 and reports an average uncertainty of >100ppm for $m^{96}\text{Zr}$, whereas the authors here often report <10ppm uncertainties. Given this stark increase in precision, the authors may want to measure standards doped with variable concentrations of Hf and verify at what threshold doubly charged Hf starts affecting the Zr isotope compositions.

We thank the referee for this constructive suggestion. We have scrutinized the Zr/Hf ratios in unprocessed aliquots of the leachates, residues, and bulk meteorite samples analyzed in this study, and all are >30. To further test the potential impact of Hf interferences, we conducted doping experiments with solutions spanning Zr/Hf ratios from 34 down to 10, i.e., a factor of ~3 lower than the least favorable Zr/Hf ratio observed in our samples. The doped solutions yielded $\mu^{96}\text{Zr}$ values indistinguishable from the undoped reference solution within uncertainty, demonstrating that the production of doubly charged Hf species is negligible at the precision of our measurements and does not compromise data quality. We now present these results in a new supplementary figure (Fig. S4) and have revised the text accordingly.

L597: Were measurement solutions checked for residual Fe? Even miniscule amounts of Fe easily compromise isotope measurements of Ru-96 due to interferences from $^{56}\text{Fe}^{40}\text{Ar}$, and the even less abundant isotope Zr-96 should be at least similarly sensitive. This is especially significant given the usage of high-sensitivity Jet and X cones in the present study, which are known to enhance the production of polyatomic species.

We thank the referee for raising this important point. We are aware that polyatomic species such as $^{56}\text{Fe}^{40}\text{Ar}$ can contribute to isobaric interference at mass 96, particularly when using Jet and X cones. To assess this, we routinely monitor the production of $^{56}\text{Fe}^{40}\text{Ar}$ on our instrument and find that it is consistently less than ~10 ppm—i.e., a 1 V signal on ^{56}Fe corresponds to a $\sim 10^{-5}$ V signal at mass 96. In addition, we routinely measure the residual Fe content in our purified Zr solutions by quadrupole ICP-MS. The resulting Fe/Zr ratios are typically better than 0.001, and often significantly lower. This high level of purification, combined with the low formation rate of $^{56}\text{Fe}^{40}\text{Ar}$, ensures that any interference at mass 96 is negligible even at the highest level of uncertainty achieved in our $\mu^{96}\text{Zr}$ measurements. This has now been clarified in the revised Methods section.

L591: ‘negligible’

Corrected.

L598: What Zr concentrations were the solutions measured at?

All solutions were measured at Zr concentrations between 25 and 100 ppb, depending on the total amount of Zr recovered for each sample. We now state this in the Methods section.

L629: Could clarify that the chemical compositions were measured from aliquots that were taken of the leachates (and that these aliquots were diluted).

We have added this information.

Figure 1: The caption reads m91Zr versus m96Zr, however, the figure shows m96Zr (y axis) plotted against m91Zr (x axis). Significant digits of the intercepts need adjustment.

Corrected.

I regressed the leachate and residue data from Table S1 using the online version of IsoplotR, which resulted in an almost identical slope (16.19 ± 0.12) and intercept (46 ± 17) for the m91Zr-m96Zr regression, however the m92Zr-m96Zr slope (42.55 ± 0.35) and intercept (2 ± 37) appear to be different. Presumably, this is related to the choice of settings for regressing the data, highlighting that the authors may want to provide more details for how these data were regressed. Finally, the χ^2 of both regressions are rather high at 11 and 5 (i.e., significant scatter in the data beyond analytical uncertainties), which could indicate an analytical artifact in the leachate data or, alternatively, the presence of a second isotopically anomalous carrier involved in generating the Zr isotope heterogeneity?

We thank the referee for recalculating the regressions and for raising this point. We have re-evaluated the regressions for both the ^{91}Zr - ^{96}Zr and ^{92}Zr - ^{96}Zr plots using either the quoted 2SE uncertainties or the external reproducibility—whichever was larger. Our recalculated slope (16.2 ± 0.1) and intercept (47 ± 18) for the ^{91}Zr - ^{96}Zr regression are identical to those obtained by the referee. For the ^{92}Zr - ^{96}Zr regression, we get the same slope and intercept as the referee if we do not consider the external reproducibility. However, we believe that it is more appropriate for these calculations to consider the external reproducibility, which yields a slope and intercept of 40.7 ± 0.9 and 52 ± 54 , respectively for the ^{92}Zr - ^{96}Zr regression. We have now updated the figure caption accordingly. We emphasize that these regressions are used solely for comparison with earlier studies and do not impact any interpretations or conclusions presented in the manuscript. Regarding the mean square weighted deviations (MSWDs), we obtain values of 8.6 for ^{91}Zr - ^{96}Zr regression and 1.8 for the ^{92}Zr - ^{96}Zr regression, indicating some scatter beyond analytical uncertainty, particularly in the former. While we cannot fully rule out minor analytical artifacts, the high precision and internal

consistency of the leachate data support their integrity. Alternatively, as the referee notes, the excess scatter may reflect the presence of more than one isotopically anomalous carrier phase contributing to the observed variability—a possibility that we consider plausible and worth exploring in future work.

Figure 2: Is the red pattern (labelled as ‘CCSN model’) referring to the He-shell of a 25M CCSN (L176-187)? Given that the nucleosynthesis of Zr is sensitive to neutron densities, does this pattern change fundamentally when a 15M CCSN is modelled instead? It would be much appreciated if the supplement would provide a table with values for these different presolar components so that the interested reader can reproduce and compare compositions and slopes.

The red pattern labelled “CCSN model” in Figure 2 refers to Zr isotopic yields from the He-shell of a 25 M core-collapse supernova, as described in the main text and in the Figure caption. We agree that the nucleosynthesis of Zr isotopes is highly sensitive to neutron density conditions, which are in turn strongly dependent on progenitor mass. In fact, we find that Zr isotopic patterns predicted for a 15 M CCSN differ significantly from those of a 25 M progenitor, both in terms of absolute abundances and relative mass-dependent trends. For this reason, the comparison in Figure 2 is intended to be more qualitative, highlighting the general shape and direction of the isotopic anomalies rather than serving as a direct fit to the data. Because of this qualitative framing, we have chosen not to include a numerical table of model abundances and presolar grain data in the Supplementary Information. We feel this could suggest a level of precision and reproducibility not warranted by the nature of the model–data comparison.

Figure 4: Caption should read FeO (y axis) as a function of $m_{96}\text{Zr}$ (x axis) as opposed to the other way around (alternatively, you could switch the two axes, so that the vertical axis is consistently $m_{96}\text{Zr}$ throughout the paper). It would be good to clarify here that the data points are the mantles of Earth, Mars, etc. (and use BSE, BSM, etc. for the labels in the figure).

We have corrected the caption and modified the labels.

Supplementary:

L23: content -> concentration

Corrected.

L123: In addition to the discrepancies between CO3 chondrites, Akram+2015 also report significantly lower $m_{96}\text{Zr}$ values for the CM chondrite group compared to the present study, as well as isotope heterogeneity between different CM2 specimen. Moreover, Table S1 reports differing $m_{96}\text{Zr}$ anomalies for the three Orgueil digestions (27.8 ± 2.2 , 38.2 ± 2 , and

33.8±3.2), which together yield a χ^2 of 24. Based on these combined observations, it seems that Zr isotopic heterogeneity is a common feature in carbonaceous chondrites, which is consistent with their unequilibrated character and the variable distribution of isotopically anomalous phases, and this would be particularly evident/problematic if only small amounts of material were dissolved. Looking at the Render+2022 reference, these authors list only a single measurement for their Orgueil sample, suggesting they indeed may have dissolved a small amount of that meteorite?

We agree that ^{96}Zr variability has been reported in previous studies for carbonaceous chondrites, including both CM and CO groups (e.g., Akram et al., 2015), and that this could reflect the heterogeneous distribution of isotopically anomalous carrier phases in these unequilibrated meteorites. Alternatively, some of the reported variability may also result from analytical artifacts. At present, it is not possible to unambiguously distinguish between these two explanations. Note that considering the external reproducibility of the ^{96}Zr measurements as opposed to the internal error, our three Orgueil replicate analyses are within analytical uncertainty. We still maintain that the offset in the ^{96}Zr value of Orgueil published in Render et al. is most likely due to the fact that these have not fully dissolved the SiC grains given that the sample was not bomb digested.

L133: 9 missing in m91Zr

Corrected.

Figure S1 and S2: It appears these correlations would be even better defined if the two NCs (NWA 5697 and NWA 753) were excluded from the regression. Perhaps they could be displayed in a different colour?

Indeed, the correlations in Figures S1 and S2 appear tighter when the two NC chondrites (NWA 5697 and NWA 753) are excluded from the regression. However, our intention was to show the full spread of our dataset and to treat all samples consistently in the visual presentation. For this reason, we have chosen not to differentiate these points by colour or symbol.

Table S1: Perhaps the authors can add a sentence or two to elaborate on how uncertainties were determined. Specifically, the uncertainties for NWA 7655 L3 (two measurements) appear a tad irregular with 2.4ppm m91Zr and 15ppm m92Zr.

For each sample, uncertainties are reported as 2SE and reflect the standard error of the mean from multiple replicate measurements. We have added a sentence to the Methods section to clarify how these uncertainties are calculated

Table S1: 'residue' and 'Roosevelt'

Corrected.

Reviewer #2 (Remarks to the Author):

Summary:

The authors conducted Zr isotope leaching and bulk measurements of chondrites to test the hypothesis that interstellar ices carried supernova produced nuclides. The authors present a unique data set, a convincing case that aqueous alteration minerals harbor extreme ^{96}Zr enrichments (a supernova signature), and that Zr was carried by interstellar ices. The methods are sound and the work supports the conclusions. The data and conclusions of this manuscript are compelling and will be of interest to the community. I congratulate the authors on a nice piece of work.

We thank the referee for the kind words and the constructive comments.

Comments:

My comments are minor and can be addressed with clarifying the text. I have no major concerns or issues with the manuscript.

L72. Recommend listing all the sample names followed by their classifications. Listing only the names of Tagish Lake, the ordinary chondrite, and the Rumuruti chondrite seems incomplete.

Agreed. We have reformulated the sentence and now include the classification.

L114: Recommend providing classifications of these meteorites at first use to make it more accessible. The above discusses CMs, and while Maribo and Murchison are CMs, Tagish Lake is not a CM.

Agreed and modified accordingly.

L123: Table 1? Should this be Table S1?

Thanks for spotting this – now corrected.

L136–138: “Since phyllosilicates are hydrous and form through the interaction of water/ice with preexisting silicates, the anomalous Zr may reflect the composition of the preexisting silicates rather than the water/ice. If correct, the isotopic composition of the anomalous Zr leached out of the phyllosilicates should match the bulk chondrite composition, which is not observed.”

-Agreed, and I agree the anomalous Zr is not from the silicates. However, the leachate should be a mixture of the silicate Zr isotope composition and that carried by the water. This is mentioned in Lines 158–161, but perhaps this statement could be brought up earlier?

Agreed, we now make this point in the section L136-138 as requested by the referee.

L144: “The most pristine samples investigated here are the two CM (Maribo and Murchison) and the Tagish Lake carbonaceous chondrites, which”

-Define what is meant by pristine, this is a relative word that is used to mean many different things (i.e., thermal alteration, aqueous alteration). Do the authors mean pristine regarding peak thermal alteration temperature?

We mean the peak thermal alteration temperature and we have now specified this as requested by the referee.

Line 296–298: “This requires a relatively constant water ice/rock ratio for the material precursor to asteroidal bodies and, moreover, indicates that thermal processing of primary solids also occurred with similar interstellar ice abundances.”

-Throughout the manuscript, there are instances where the word ice is used, but I think what is meant is the interstellar ice component. This can be clarified in the text. For example, I think the first use of ice in the above sentence is intended to be interstellar-ice/rock ratio, not total ice.

We agree with the referee and have corrected this accordingly where appropriate.

Figure 4. Recommend the figure label should read ‘interstellar ices’, not just ice. Or was total water/ice meant here?

Agreed and corrected to interstellar ices.

Supplementary File:

Table S1.

-Roosevelt is misspelled, it should be Roosevelt.

Corrected.

The authors have effectively addressed previous comments and have included helpful clarifications to accommodate potential concerns, leading to significant improvements in the revised version of the manuscript. I have a few more minor comments (mostly clarifications related to the new paragraph starting in line 43), however, I do not need to see another revision of the manuscript.

Thanks again for reading our paper carefully.

Minor comments:

L32: Suggest adding 'primarily' before 'imparted'

Done.

L55: Replace 'outer' with 'outermost' to clarify that the comet region is meant and not the CC region

Done.

L59: Suggest adding 'inferred to be' before 'from the comet-forming region'

Done.

L60: It is mentioned later in the manuscript; however, it may be good to clarify here that these are (not Zr but) combined Mg-Si-Fe-Cr nucleosynthetic isotopic signatures

Done.

L62: Yes, however, more accurately: "[...] in tension with models that invoke a CAI-like isotopic composition of early accreted envelope materials."

Done.

L218: Something missing in this sentence ("in the leachates"?)

Thanks. Yes, leachates. Corrected.

L247: Suggest adding 'the thermal processing' (or 'the selective incorporation') before 'of interstellar ices'

Done.